# Mathematical basis for the assessment of antibiotic resistance and administrative counter-strategies

Hans H. Diebner [1¤]*, Anna Kather[2], Ingo Roeder[1]*, Katja de With[2]

1 Carl Gustav Carus Faculty of Medicine, Institute for Medical Informatics and Biometry, Technische Universität Dresden, Dresden, Germany, 2 University Hospital Carl Gustav Carus Dresden at the TU Dresden, Division of Infectious Diseases, Dresden, Germany

¤ Current address: Department of Medical Informatics, Biometry and Epidemiology, Ruhr-Universität Bochum, Bochum, Germany

* hans.diebner@tu-dresden.de (HHD); ingo.roeder@tu-dresden.de (IR)

**Data Availability Statement:** All relevant data are within the manuscript and its Supporting Information files.

## Abstract

Diversity as well as temporal and spatial changes of the proportional abundances of different antibiotics (cycling, mixing or combinations thereof) have been hypothesised to be an effective administrative control strategy in hospitals to reduce the prevalence of antibiotic-resistant pathogens in nosocomial or community-acquired infections. However, a rigorous assessment of the efficacy of these control strategies is lacking. The main purpose here is to present a mathematical framework for the assessment of control stategies from a processual stance. To this end, we adopt diverse measures of heterogeneity and diversity of proportional abundances based on the concept of entropy from other fields and adapt them to the needs in assessing the impact of variations in antibiotic consumption on antibiotic resistance. Thereby, we derive a family of diversity measures whose members exhibit different degrees of complexity. Most important, we extent these measures such that they account for the assessment of temporal changes in heterogeneity including otherwise undetected diversity-invariant permutations of antibiotics consumption and prevalence of resistant pathogens. We apply a correlation analysis for the assessment of associations between changes of heterogeneities on the antibiotics and on the pathogen side. As a showcase, which serves as a proof-of-principle, we apply the derived methods to records of antibiotic consumption and prevalence of antibiotic-resistant germs from University Hospital Dresden (cf. supplement "DiebnerEtAl_Data-Supplement"). Besides the quantification of heterogeneities of antibiotics consumption and antibiotic resistance, we show that a reduction of prevalence of antibiotic-resistant germs correlates with a temporal change of similarity with respect to the first observation of antibiotics consumption, although heterogeneity remains approximately constant. Although an interventional study is pending, our mathematical framework turns out to be a viable concept for the assessment and optimisation of control strategies intended to reduce antibiotic resistance.

**Funding:** Open Access Funding by the Publication Fund of the TU Dresden.

**Competing interests:** The authors have declared that no competing interests exist.

## Introduction

The drastic increase of antimicrobial resistance worldwide resulting in an alarming increase in morbidity and mortality from clinical infections urges scientists and clinicians to develop counter-strategies [1]. Designing new antiinfective agents is an option. However, the creation of new drugs is time-consuming and success is not guaranteed. Therefore, the control of consumption of available antibiotics or, more general, of antiinfectives, is obligatory.

Since a quick replacement of existing antibiotics is not feasible, antibiotic strategies as "antibiotic mixing", "antibiotic cycling", "antibiotic switching", and "rotation protocols" gained evermore attention in the recent decades [2–6] (for a review cf. [7]). All these concepts refer to heterogeneity of antibiotic usage and non-constant prescription rates. For example, antibiotic cycling means to extract one or a subset of classes of antibiotics from administration in a temporarily alternating way whereas other strategies refer to a scheduled change of the dominantly used class of antibiotics. Frequently, mixing refers to a strategy where a group of patients receives drug (class) A and another group receives drug (class) B in an alternating way. Hereby, the permutation usually takes place between wards leading to a spatial heterogeneity of antibiotic consumption. In clinical reality, empirical antibiotic therapy rarely follows strictly defined control schemes but rather adjusted forms of cycling and mixing in consequence of clinical requirements, thus "clinical cycling" (notion adopted from [4]).

Although there is some evidence that the temporal and spatial permutations of rates of consumption of different antiinfectives are able to reduce prevalence of resistant pathogens [2–4, 8], there is a lack of rigorous evaluations of the ongoing processes, which prevents optimisation. The nonlinear time series analysis presented by Lopez-Lozano et al. 2019 is indicative for an approach from a processual stance, but needs to be generalised. Usually, the impact of cycling and mixing strategies (for a review and meta analysis cf. [7]) is investigated by means of prospective (randomized clinical trials (RCTs), controlled clinical trials (CCT), controlled before-after, cross-over) based on well-defined rotation/switch protocols, thereby neglecting (continuous real-world) processes. Interrupted time series analyses with at least three observations before and after intervention exist, but once more, a precise time point of intervention is preconditioned, thereby excluding adjusted strategies of cycling and mixing. Of note, other researchers as e.g. Karam et al. [5] could not confirm a significant reduction of antimicrobial resistance through cycling or mixing protocols.

The main purpose of this work is to present an analytical framework to quantify heterogeneity of both, antibiotic consumption as well as prevalence of antibiotic-resistant pathogens as a function of observation time. This enables the assessment of associations between consumption and resistance and, potentially, to optimise control strategies based on diversity, mixing and cycling either in their adjusted versions, i.e. "semi-controlled" field-like or observational studies, or in their extremes. The analytic framework consists of adapted methods known in other areas like ecosystems [9, 10]. For the purpose of illustrating the mathematical framework, the method is applied to real observational data, i.e., to records of antibiotic consumption and prevalences of antibiotic-resistant pathogens of University Hospital Dresden in Germany (cf. supplement "S1 File"). Although these data have been recorded in the absence of a clearcut study design and without genuine applications of common cycling or mixing strategies, we nevertheless opted to use these data instead of simulated data for illustration purpose and to preserve a "real world" character of our methodological approach. Thus, the application of the proposed mathematical framework has to be understood as a proof-of-principle. Rigorous controlled interventional studies are planned, however, we regard the immediate provision of the analytical framework to have utmost priority.

## Materials and methods

### Records of antibiotic consumption

The available data set (cf. supplement "S1 File") contains 25 consecutive quarterly records of antibiotic consumption starting from the first quarter 2012. Consumption has been recorded per administrative units (cost centres, typically wards). However, for the elementary stage of illustrating the proposed method, a grouping of these small subunits into functional units is considered to be sufficient. Therefore, the departments are grouped into the three top-level units: surgical/OP units, intensive care units, and medical/normal care units. Consumption has also been recorded per active agent, which is nested within antibiotic class. In total, 49 active agents have been observed, which are pooled to 12 antibiotic classes.

The consumption of an antibiotic is measured in standardised units according to the ATC/WHO definition of defined daily doses, $DDD$, in order to allow for comparisons of different active agents. In addition, the number of cases as well as patient days have been recorded on a quarterly basis. This allows to compute the consumption density $DDD$ per 100 patient days in hospitals:

$$DDD_{density} = \frac{DDD}{100 \text{ patient days}}.$$ (1)

Please note, for the sake of completeness, consumption density sometimes refers to $DDD$ per 100 or per 1000 cases, respectively. In the following, we use $DDD$ only since $DDD_{density}$ gives virtually the same results (data not shown). Moreover, some measures of diversity are functions of proportions of "species" within an "ecosystem", which is why we make use of proportions of consumption. If $DDD_i(t)$ denotes the consumption of antibiotics within the antibiotic class $i \in \{1, \ldots, n\}$ at time $t$, the proportion is given by

$$ddd_i(t) = \frac{DDD_i(t)}{\sum\limits_{i=1}^{n} DDD_i(t)}$$ (2)

Depending on the context, index $i$ may also refer to the active agent.

### Coefficient of variation

The coefficient of variation,

$$V(t) = \frac{SD(DDD(t))}{Mean(DDD(t))},$$ (3)

where the mean is taken over the antibiotic classes and $SD(DDD(t))$ denotes the corresponding standard deviation, can be used as a rough estimate of "homogeneity" in a properly defined sense. Unfortunately, analogous to the notion of "dispersion," "homogeneity" is an ambiguous term which deserves clarification. In ecological analyses, the concept of "maximisation of statistical heterogeneity" refers to an approach by means of an entropy or a related diversity measure as discussed in the following section (cf. [11]). In this context, an ecosystem is maximally heterogeneous, thus has maximum entropy, if all species are equally abundant, at least in the absence of specific weights of the species abundances [9]. In the latter case, $V$ would be zero, i.e., the system has no variability and is thus without (statistical) dispersion. In contrast, physicists prefer to speak of a perfect dispersion, aka a perfect mixture, in such a situation. Thus, an ecosystem or a society close to a monoculture is "homogeneous" whereas a multi-cultural society/ecosystem is called "heterogeneous" which is used synonymously to "diversity."

In economics, to the contrary, the distribution of incomes is called homogeneous in the case of equal incomes of all individuals in accordance with the idea of a dispersion-free perfect mixture [12]. Due to compatibility, we stick with the ecological approach in the sequel. In this regard, the coefficient of variation is an inverse measure of ecological heterogeneity. Arguably, heterogeneity in the ecological sense is better captured using the concept of diversity, as introduced in the following section. In order to provide compatibility with the terminology based on the notion of "heterogeneity" suggested in relevant publications on antibiotics resistance [2, 13], we cannot completely drop this term.

## Heterogeneity and entropy

Although the coefficient of variation can be interpreted as a rough measure of (inverse) heterogeneity, it has several inadequacies including the lack of uniquely capturing temporal changes. In the sequel, we harness methods known in ecological population modelling and other fields of research for an adequate quantification and assessment of antibiotic mixing behaviour and strategies as well as temporal patterns of prevalence of antibiotic resistance.

Let $a_i$ and $b_i$ with $i = 1, \ldots, n$ be the proportions of species of two $n$-species populations composed of the same set of species with $\sum_{i=1}^{n} a_i = \sum_{i=1}^{n} b_i = 1$. Similarity of these two populations in terms of species' abundances can be quantified by the similarity index

$$SI = 1 - 0.5 \sum_{i=1}^{n} |a_i - b_i|. \tag{4}$$

If $a_i = b_i, \forall i = 1, \ldots, n$, then $SI = 1$, i.e., the populations are identical in terms of their species distributions. If, on the contrary, the populations consist of disjoint sets of species, then $SI = 0$. Similarity index $SI$ scales between 0 and 1. If we now fix say the first population to $a_i = \frac{1}{n}, \forall i = 1, \ldots, n$, which means maximum heterogeneity for this reference population, then, for the other population a heterogeneity index can be defined by

$$HI = 1 - \frac{n}{2(n-1)} \sum_{i=1}^{n} \left| \frac{1}{n} - b_i \right|. \tag{5}$$

Hereby, the slightly adapted factor $\frac{n}{2(n-1)}$ compared to 0.5 in $SI$ (Eq 4) ensures $HI \in [0, 1]$ independent from the concrete value of $n$. As far as we know, $HI$ defined by Eq 5 has been used by Sandiumenge et al. [2] for the first time in the context of assessing antibiotic resistance and reapplied by Plüss-Suard et al. [13]. Please note that the typesettings of the formulas for $HI$ in refs. [2, 13] are incorrect. Abel zur Wiesch and collaborators [4] explicitly call this measure "antibiotic heterogeneity index (AHI)", although it originated in other fields of research.

Diversity, a notion frequently used in ecology, is a more general concept than heterogeneity [9, 14]. However, diversity is not uniquely defined. Thus, it depends on the specific context to which particular definition of diversity should be drawn on. Even within the field of antibiotic consumption and related antibiotic resistance, the concrete context can vary considerably. Since we here aim at presenting a general mathematical framework, a family of measures whose members are characterised by exhibiting different levels of complexity is presented.

To start with, an obvious somewhat simplistic way to quantify diversity is given by the so called richness, which is merely the number of species in a multi-species population (e.g. ecosystem). In terms of richness, a heterogeneous $n$-species population with equally frequent species has the same diversity as an $n$-species population with a minority of dominating and a majority of very rare species, that is to say $n$. It follows that an expedient diversity measure should be based on the distribution of species' abundances in some way.

A meaningful definition of diversity $D_a$ is based on an "effective number of species" (cf. [9, 14]) given by the species' proportions $p_i$ and a weight parameter $a$ by means of:

$$D_a = \left(\sum_{i}^{n} p_i^a\right)^{\frac{1}{1-a}}. \tag{6}$$

Setting $a = 0$ yields $D_0 = n$ independently from $p_i$, i.e. richness. Other special cases are:

$$D_1 = e^{R_1}$$

$$D_2 = \frac{1}{\sum_{i=1}^{n} p_i^2} \tag{7}$$

$$D_\infty = \frac{1}{\max(p_i)}$$

with $R_1 = -\sum_{i=1}^{n} p_i \ln(p_i)$ being the so called "Shannon entropy", sometimes also called "Shannon index." A unique name for $D_1$ itself, i.e. the exponential of the Shannon entropy, does not exist, however, in information science it is sometimes called "perplexity." Diversity measure $D_2$ is called "inverse Simpson index." The frequently used Gini-Simpson-Index derived from $D_2$ is given by: $GS = 1 - \frac{1}{D_2}$. The inverse of $D_\infty$ is found in the literature named "Berger–Parker index," which is simply the proportional abundance of the most abundant type.

The Shannon entropy is a special case of a Renyi entropy defined by

$$R_a = \frac{1}{1-a} \ln\left(\sum_{i=1}^{n} p_i\right), \tag{8}$$

thus we have $D_a = e^{R_a}$. In other words, $R_a$ is a monotonous function of $D_a$, thus, the two measures can be used interchangeably without loss of information since diversity has only a relative meaning, anyway. The same holds for $GS$ and other possible monotonous functions of $D_a$. Entropy $R_a$, thus $D_a$, independently of $a$ reach their maximum for the fully heterogeneous situation $p_i = \frac{1}{n} (\forall i = 1, \ldots, n)$ and it then follows that $D_a = n$. From the latter result we conclude that richness might be a sufficient diversity measure for populations close to full heterogeneity. Having said that, heterogeneity $HI$ itself, although it cannot be derived as a special case of a Renyi entropy, shares features of an entropy and is thus a legitimate measure of diversity.

Finally, the frequently used Gini coefficient deserves to be mentioned:

$$G = 1 - \frac{1}{2(n-1)} \sum_{i=1}^{n} \sum_{j=1}^{n} |p_i - p_j|. \tag{9}$$

The Gini coefficient in this form (for this and other variants see [14]) is an interesting variant of the similarity index $SI$ insofar as it can be interpreted as kind of a self-similarity. Once more, full heterogeneity (equal proportions) $p_i = \frac{1}{n}, \forall i = 1, \ldots, n$, implies $G = 1$, and maximally unequal proportions (e.g. $p_i = 1, p_{j \neq i} = 0$) implies $G = 0$.

For what follows, it is important to bring to mind that both heterogeneity, $HI$, as well as measures of diversity, $D_a$, $GS$, and $G$, are invariant under permutations of indices. In other words, if the proportions of two species are exchanged, diversity (heterogeneity) does not

change. However, similarity index *SI* is capable to account for such a change after some adaptations, as shown in the Results section. In order to assess both (temporal) cycling as well as (spatial) mixing behaviour of antibiotic administration or consumption, respectively, the proper similarity index extends to

$$SI = 1 - 0.25 \sum_{i=1}^{n} \sum_{j=1}^{m} |a_{ij} - b_{ij}| \tag{10}$$

where $\sum_{i=1}^{n} |a_{ij}| = \sum_{i=1}^{n} |b_{ij}| = 1 \ \forall \ j = 1 \ldots m$. Hereby, the inner sum is taken over the $m$ wards or groups of patients between which antibiotic mixing takes place. In this case, $a_{ij}$ and $b_{ij}$ refer to the relative abundances before and after the swap of antibiotics administrations between the wards, respectively. In practice, i.e. in the absence of a properly defined study protocol, an allocation of both antibiotic consumption as well as the prevalence of resistant pathogens to precisely defined wards or groups of patients is hampered by the reality of adjusted clinical cycling/mixing. In the following, due to lack of appropriate information on mixing strategies, we present definitions of similarity tailored to our needs without taking strict mixing into account.

## Statistical analysis

Main purpose of this article is to apply a diversity analysis to records of both clinical consumption of antibiotics as well as prevalence of pathogens exhibiting antimicrobial resistance. We refer to the previous section for a detailed introduction of the applied diversity measures. It remains to mention that the expected impact of diversity of antimicrobial consumption on the prevalence of resistant pathogens is analysed by means of Pearson's correlations of the corresponding time series.

Specifically, slopes along with their significance of differing from zero taken from linear regression quantify the temporal changes of both diversity as well as differential diversity. Of particular interest is the comparison between the time courses of differential diversities of antibiotic consumption and prevalence of resistant pathogens. Such a comparison can be achieved by testing of whether the two corresponding slopes differ significantly or not, or, equivalently, by calculating Pearson's correlation coefficient along with the corresponding significance test. In the same line, the time series of the relative abundance of resistant pathogens is tested for correlations with the time course of the differential diversity of antibiotic consumption using, as before, Pearson's product moment correlation analysis.

Of note, each of the three time series, i.e. differential diversity of consumption, differential diversity of prevalence, and relative abundance of resistant pathogens, are expected to exhibit autocorrelations. This is a common feature of time series analyses. Nevertheless, Pearson's correlation is commonly used as a standard technique to quantify the co-variation of two correlated time series and we here regard it to be sufficient for raising hypotheses based on the available preliminary dataset. A more appropriate application of a cross-correlation function with one of two time series being subject to a time lag is not applicable in our case due to the insufficient lengths (and low sampling frequencies) of the time series.

Numerical calculations, statistics, and graphics have been performed with R [15].

## Results

### Preliminary note

Before presenting the results of applying the derived diversity measures to observational clinical data, we reemphasise that this work mainly aims in presenting the mathematical

framework. The following application to real data has an illustrative purpose and outlines future applications to adequately collected data from a controlled trial. Therefore, in terms of clinically relevant results, the following explanations remain provisional.

## Descriptive analysis

Fig 1a shows the 12 time courses of antibiotic consumption per antibiotic class, $DDD_i(t)$. The consumptions of 9 classes largely remain constant on a moderate level. One class, that is to say "second-generation cephalosporins", is characterised by a high consumption at the outset but declines approximately monotonously by more than half towards the end of the observation period. The consumptions of two other classes, in contrast, i.e. "aminopenicillin/beta-lacta-mase inhibitors" and "narrow-spectrum penicillins" increase approximately monotonously and roughly compensate for the aforementioned decline.

Calculations based on $DDD$ are hardly distinguishable from calculations based on the corresponding densities, $100 \times DDD_i$/patient days. Henceforth, due to these minor differences,

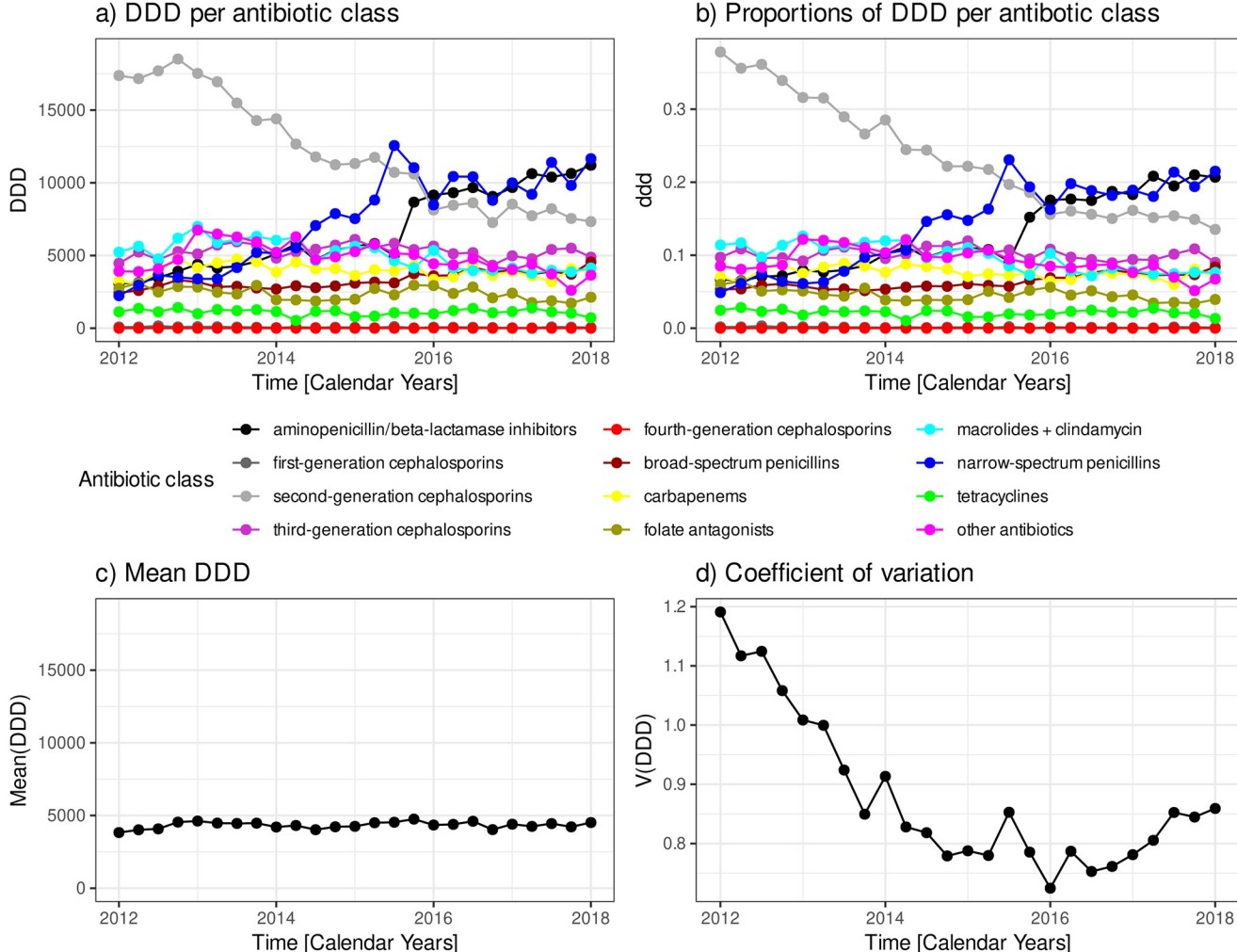

**Fig 1. Time course of antibiotic consumption by antibiotic class.** Time courses of a) consumption DDD per antibiotic class, b) proportions of consumption ddd per antibiotic class, c) mean consumption averaged over the antibiotic class, d) coefficient of variation with respect to the antibiotic classes.

we skip to report our results with respect to consumption densities since we here primarily deal with an introduction of a methodological concept. The distinction between DDD and the corresponding densities might become important in other contexts, though. The proportions, $ddd_i(t)$, however, depicted in Fig 1b, will be used in later sections where we calculate measures of diversity.

Fig 1c shows the time course of the quarterly sampled mean antibiotic consumption, *Mean* ($DDD(t)$), averaged over the 12 observed antibiotic classes (cf. Fig 1a). The corresponding coefficient of variation, Eq 3, is depicted in Fig 1d.

In the present case, variability thus homogeneity in the ecological sense is rather high during the first 4 to 6 quarters compared with the remaining time course. After an approximately monotonous decline until 2016, $V(t)$ slightly increases again during the final quarters. These results are consistent with the visual impressions from Fig 1a. Starting with a rather homogeneous distribution at the outset with an outstandingly large proportion of a single antibiotic class, we observe a trend towards a narrow distribution around the mean that starts to weakly widen towards the end. Equal proportions, thus $V(t) = 0$, means perfect heterogeneity, therefore, $V(t)$ can be interpreted as an inverse measure of heterogeneity.

In the same line, the descriptive analysis can be applied to consumption with respect to active agents. Fig 2a shows the 49 time courses of consumption per active agent. We observe that a single active agent, viz. "cefuroxime", dominates consumption at the outset but declines approximately monotonously towards the end to a level still significantly above the bulk. This decline is compensated by an increase in consumption mainly of "amoxicillin + clavulanic acid" but also some other agents. The time courses of mean DDD and the coefficient of variation with respect to the active agents shown in Fig 2b and 2c reveal that variability remains on a high level during the time course. Thus, it turns out that pooling agents into antibiotic classes has a damping effect with respect to variability or homogeneity, respectively.

Next step is to account for the Hospital's functional units. The three panels of Fig 3 show the time courses of antibiotic consumption per antibiotic class stratified by the three functional units: unit 1 = intensive care units, unit 2 = medical/normal care units, unit 3 = surgical/OP units. Unit 1 consumed antibiotics out of 11 classes, whereas unit 3 consumed antibiotics out of only 8 different classes in at least one quarter during the whole observation period. Only unit 2 has non-zero consumption of antibiotics out of all 12 classes in at least one quarter.

Fig 4 shows the time courses of mean consumption averaged over the antibiotic classes per functional unit (Fig 4a) as well as the three corresponding coefficients of variation (Fig 4b). Unit 3 (the surgical/OP units) exhibits the lowest mean consumption but by far the highest coefficient of variation, both of which remain approximately constant over the time course. The explanation follows by throwing a glance on Fig 3c: Unit 3 has one absolutely dominating consumption of antibiotics out of the class "second-generation cephalosporins." Units 1 and 2 both exhibit approximately temporarily constant mean consumption, however, unit 2 on a roughly 5-fold higher magnitude. Noteworthy, the variation of consumption of unit 2 approximately follows the variation for the whole clinic with a more or less monotonous decline during the first half of the observation period, whereas the coefficient of variation for unit 1 is approximately constant over the time course, with the exception of a marked rise at the final observation (first quarter of 2018), which can be explained by the sudden rise of consumption of "narrow-spectrum penicillins" antibiotics (cf. Fig 3a).

## Heterogeneity of antibiotic consumption

The diversity measures introduced in the "Materials and methods" section are now calculated using the observed proportions $p_i$ of antibiotic consumption with respect to antibiotic classes,

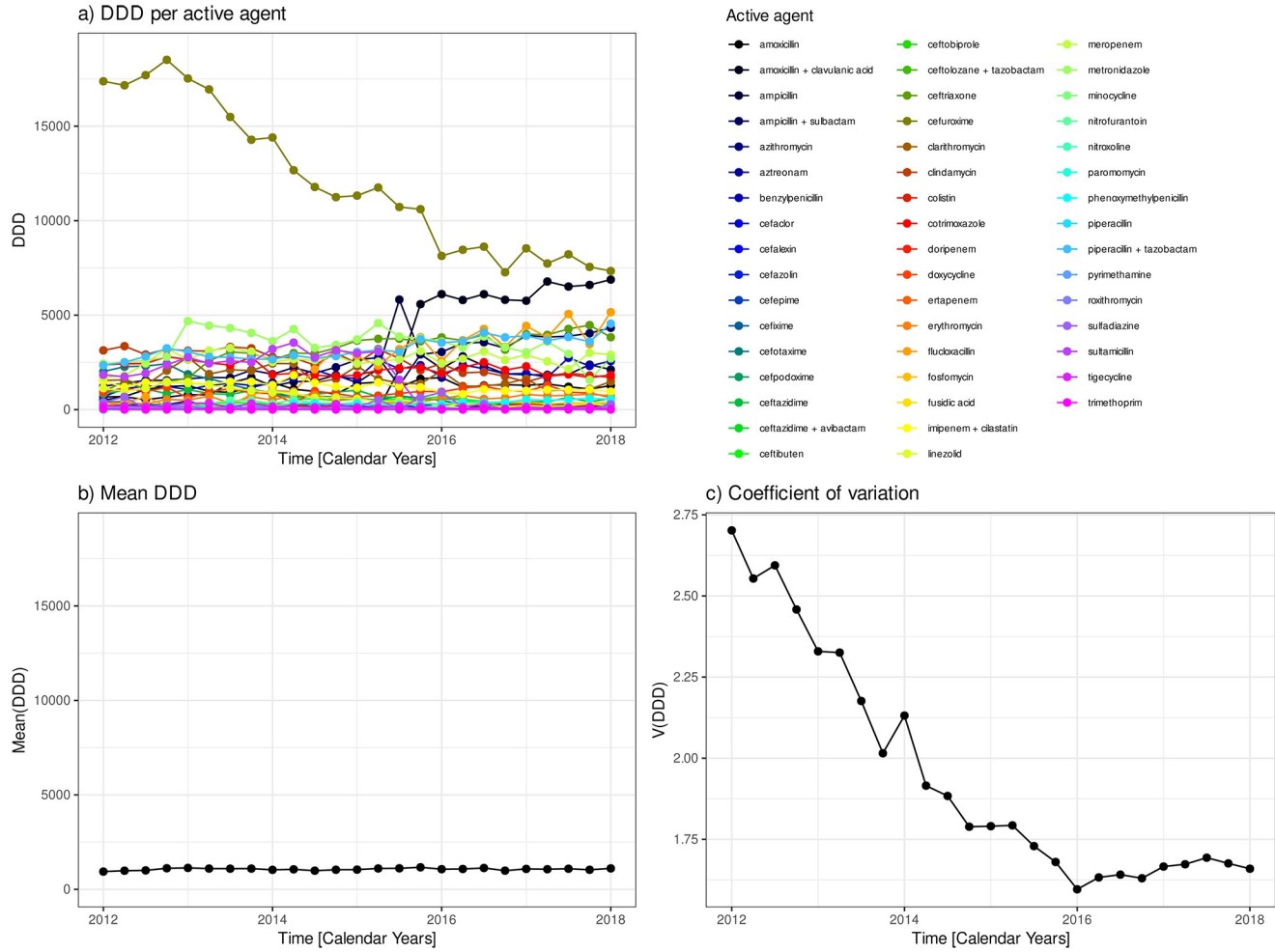

**Fig 2. Time course of antibiotic consumption by active agent.** Time courses of a) consumption DDD per active agent, b) mean consumption averaged over the active agents, c) coefficient of variation with respect to the active agents.

thus $i = 1, \ldots, 12$ refers to the 12 antibiotic classes. Fig 5 shows time courses of a) Renyi entropies, $R_a$, for 7 different values of weight parameter $a$ (cf. figure legend), b) the corresponding diversities, $D_a = e^{R_a}$ for the same set of parameters $a$, c) the Gini-Simpson diversity, $GS$, and d) the Gini coefficient, $G$. Specifically, $a = 0$ yields $D_0 = n = 12$ (Fig 5b) in agreement with what we expected from theory.

Apparently, for all $a > 0$ the curves exhibit the same shape, i.e., they differ at each time point by a factor that seems to be a monotonous function of the values of a reference curve at these time points. A unique rule which specifies the best value of $a$ does not exist. However, some researchers invoke a rule of thumb (e.g. [10]) which states that if one exactly observes such a monotonous relation between the different curves as we did, then the Renyi entropy is a robust measure and $a > 0$ can be chosen arbitrarily but consistently, since entropy does not have an absolute meaning, anyway. Apparently, the rule also holds for the Gini-Simpson diversity, $GS$, and the Gini coefficient $G$.

Fig 5 reveals that all diversity measures increase from 2012 until approximately 2016. Thereafter, this tendency is stopped and the data even suggest the initiation of a decrease in diversity. This behaviour coincides with the time course of the coefficient of variation (Fig 1).

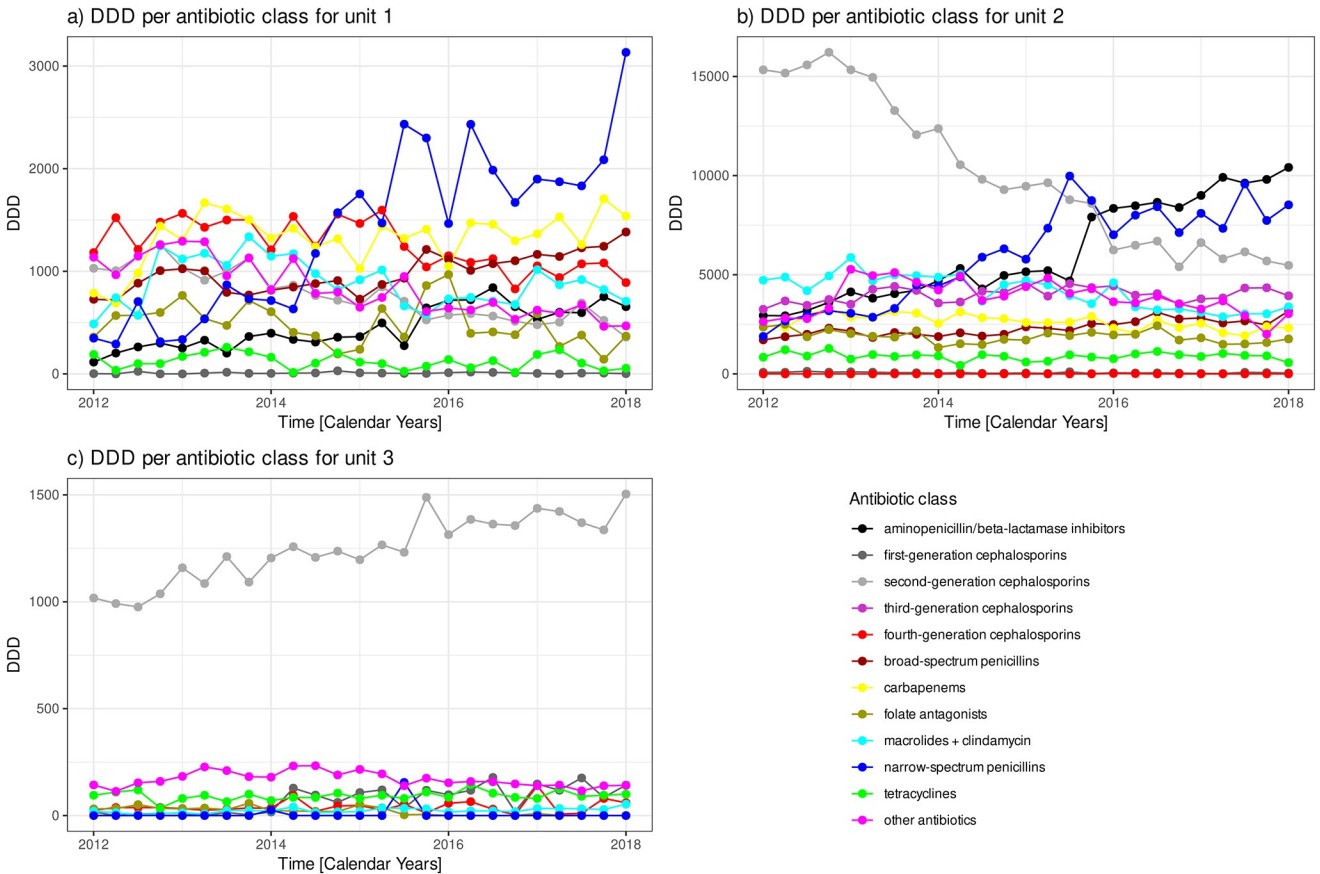

**Fig 3. Time courses of antibiotic consumption by antibiotic class separated by functional units.** Time courses of antibiotic consumption per antibiotic class shown separately for the functional units a) unit 1 = intensive care units, b) unit 2 = medical/normal care units, c) unit 3 = surgical/OP units. Please note, the different scales of the y-axes reflect the different total amounts of consumption within each unit due to their different sizes. Important are the relative abundances within each unit.

Compared to the Renyi entropy, heterogeneity *HI*, defined in Eq 5, is easier to comprehend, which might explain that its application in the context of antibiotic resistance is unparalleled up to date [2, 4, 13]. However, the invariance under species permutations has not been taken into account thus far. A cycling strategy that dictates an occasional temporal swap of consumption of two antibiotics with respect to the hospital unit under consideration leads to a temporarily constant heterogeneity and, therefore, to wrong conclusions if the assessment is merely based on *HI*. Likewise, a mixing strategy, i.e. switching the consumption of two antibiotics between two units would remain unnoticed unless the diversity of the two units are not separately calculated (cf. Eq 10). Since heterogeneity is nothing but a special case of similarity *SI*, we can now make use of *SI* to introduce a differential measure. Similarity with respect to the initial observation at the outset of a study

$$SI_0(t) = 1 - \frac{n}{2(n-1)} \sum_{i=1}^{n} \left| p_i(t=0) - p_i(t) \right|,$$  (11)

with *t* = 0, 1, . . ., 24 being the number of elapsed quarters, captures changes with respect to the

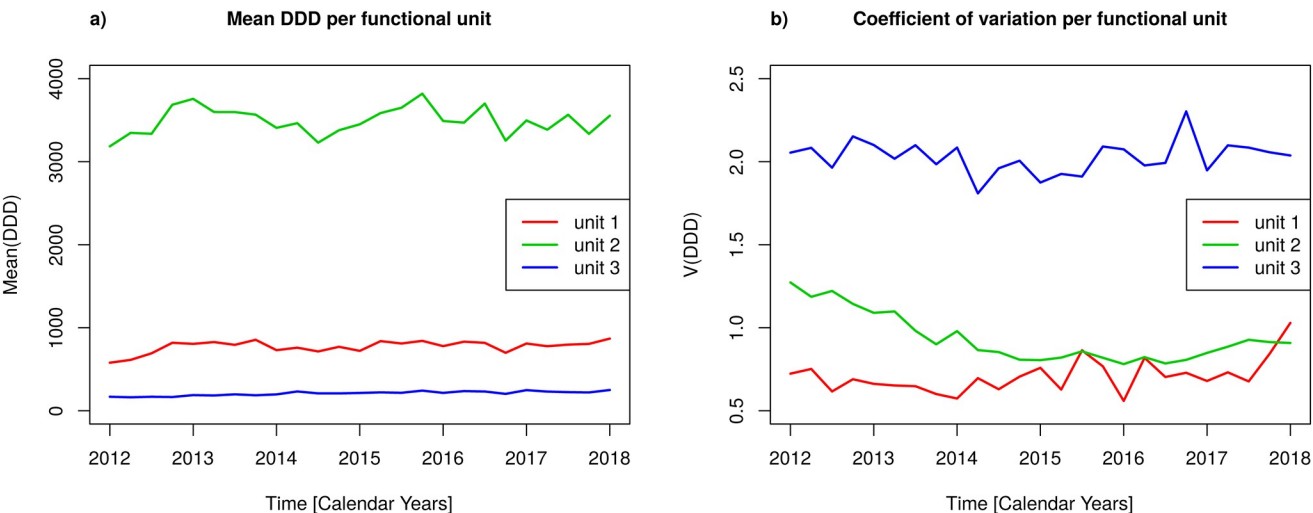

**Fig 4. Mean antibiotic consumptions and coefficient of variation by functional unit.** Time courses per functional unit of a) mean antibiotic consumption averaged over antibiotic classes, b) corresponding coefficient of variation, with unit 1 = intensive care units, unit 2 = medical/normal care units, unit 3 = surgical/OP units.

first observation. In the same line,

$$SI_\Delta(t) = 1 - \frac{n}{2(n-1)} \sum_{i=1}^{n} \left| p_i(t-1) - p_i(t) \right|, \quad \text{for } t > 0. \tag{12}$$

defines changes with respect to consecutive quarters, thus defines an approximation to a differential measure of similarity. The time courses of $HI$, $SI_0$, and $SI_\Delta$ are depicted in Fig 6a–6c. Heterogeneity $HI$ varies only within a small range from 0.63 to 0.69 (Fig 6a). However, we observe a monotonous, almost linearly increasing displacement from the first observation (Fig 6b). Due to this linear increase, the magnitude of the differential displacement $SI_\Delta(t)$ is more or less constant and is approximately one minus the slope of $SI_0(t)$ (Fig 6c). Obviously, the difference of the distribution of antibiotic abundances accumulates over the time course, where $SI_\Delta(t)$ measures the intensity of the change as a function of time. Thus, we now have a sound basis for the evaluation of changing abundances and possibly related switching strategies.

Within the scope of physics and information sciences, it is common to base measures of heterogeneity and diversity, respectively, on the Shannon entropy because it can be interpreted as the negative mean information that arises from averaging over the individual contributions $\ln(p_i)$ to information. In this context, the similarity index between two distributions given by $a_i$ and $b_i$ corresponds to the Kullback-Leibler divergence [11]

$$KLD(a_i, b_i) = \sum_{i=1}^{n} a_i \log\left(\frac{a_i}{b_i}\right). \tag{13}$$

However, $KLD$ is asymmetric, which is why the symmetric variant $KLD(a_i, b_i) + KLD(b_i, a_i)$, known as Kullback-Leibler difference, has been introduced. $KLD$ is the mean information difference taken over the individual information differences $\log(a_i) - \log(b_i)$.

In order to harness $KLD$ for our needs, we define the Kullback-Leibler heterogeneity

$$KL(t) = 1 - \frac{1}{2\ln(2)} \left( \sum_{i=1}^{n} a_i(t) \log\left(\frac{a_i(t) + 1}{\frac{1}{n} + 1}\right) + \sum_{i=1}^{n} \frac{1}{n} \log\left(\frac{\frac{1}{n} + 1}{a_i(t) + 1}\right) \right). \tag{14}$$

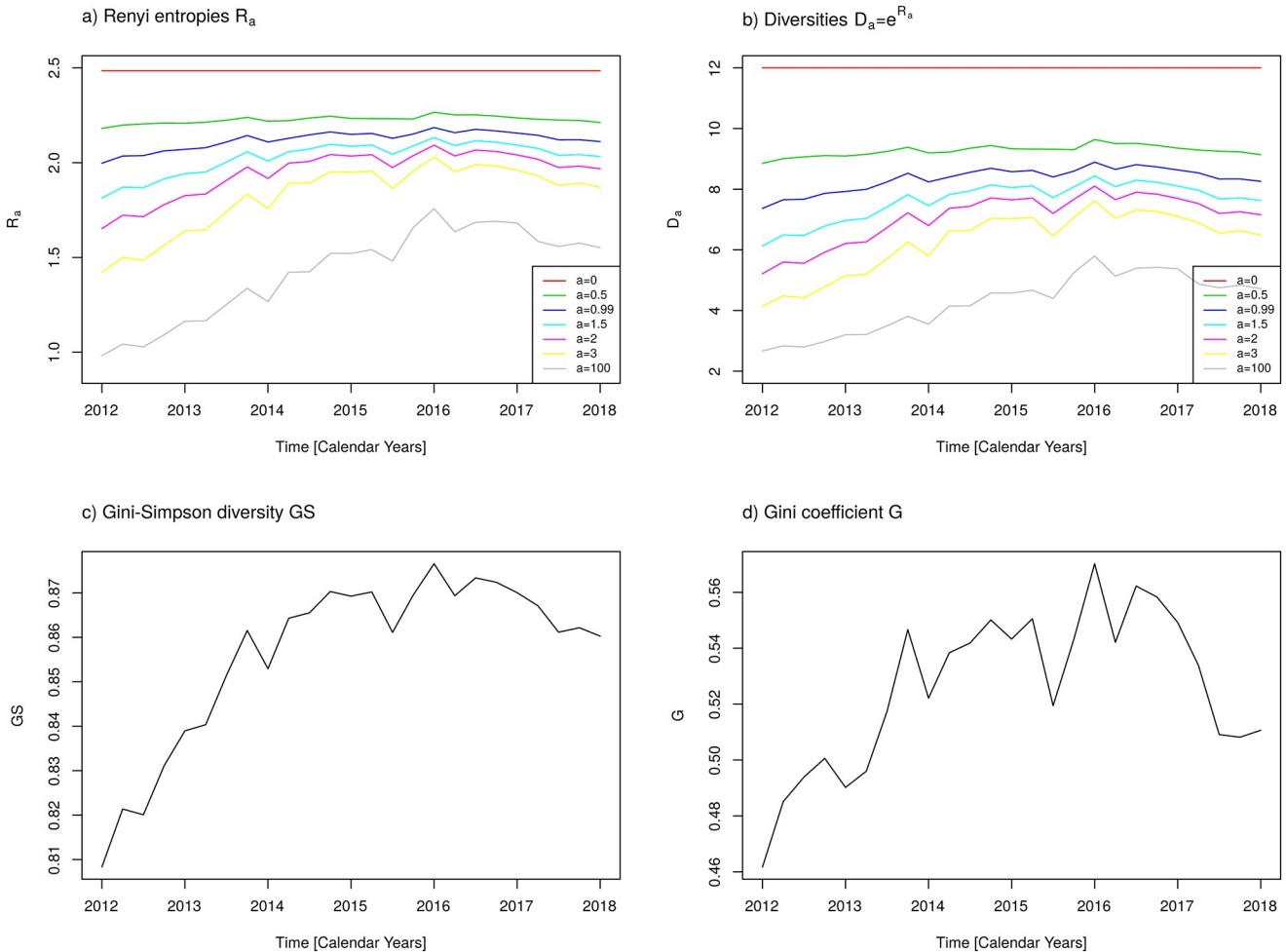

**Fig 5. Measures of diversity with respect to antibiotic classes.** a) Renyi entropies $R_a$ for $a = 0$, 0.5, 0.99, 1.5, 2, 3, 100, b) Diversities $D_a$ for $a = 0$, 0.5, 0.99, 1.5, 2, 3, 100, c) Gini-Simpson index GS, d) Gini coefficient G.

Furthermore, the Kullback-Leibler similarity $KL_0(t)$ of distribution $a_i(t)$ at time $t$ with the distribution at $t = 0$ (first observation) can be defined by

$$KL_0(t) = \sum_{i=1}^{n} a_i(t) \log \left( \frac{a_i(t) + 1}{a_i(t = 0) + 1} \right) + \sum_{i=1}^{n} a_i(t = 0) \log \left( \frac{a_i(t = 0) + 1}{a_i(t) + 1} \right). \quad (15)$$

Finally, the Kullback-Leibler similarity $KL_\Delta(t)$ between two distributions observed at subsequent time points (here quarters) is given by

$$KL_\Delta(t) = \sum_{i=1}^{n} a_i(t) \log \left( \frac{a_i(t) + 1}{a_i(t - 1) + 1} \right) + \sum_{i=1}^{n} a_i(t - 1) \log \left( \frac{a_i(t - 1) + 1}{a_i(t) + 1} \right). \quad (16)$$

Hereby, the + 1 terms within the arguments of the logarithms ensure that situations with $p_i = 0$ remain well-defined.

Fig 6d–6f show time courses of $KL$, $KL_0$ and $KL_\Delta$, respectively. A comparison with the ordinary heterogeneity and similarity measures of Fig 6a–6c reveals that the values of $KL$, $KL_0$ and $KL_\Delta$ are located within narrower intervals. In addition, $KL$ appears to be smoother than $HI$ and, noteworthy, the decreasing curve of $KL_0(t)$ has a concave shape. From these differences

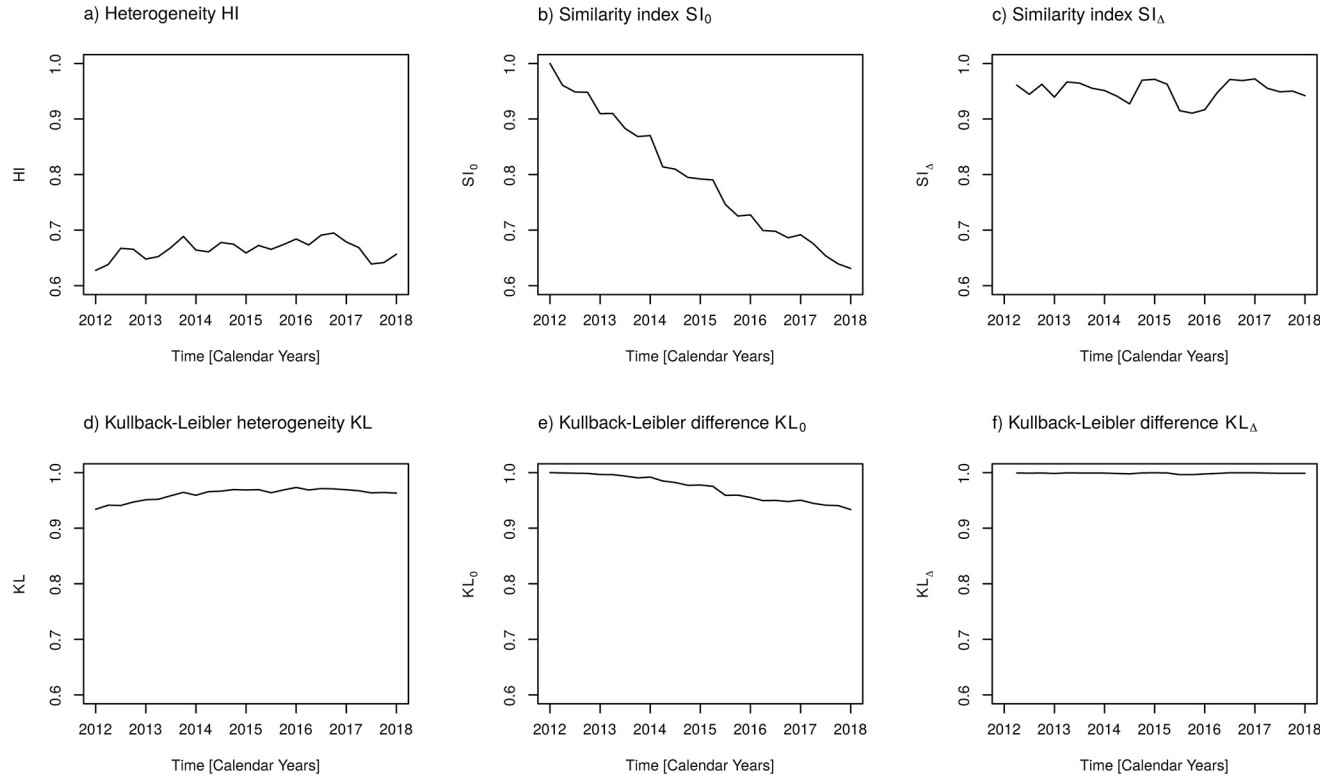

**Fig 6. Measures of heterogeneity and similarity with respect to antibiotic classes.** a) Heterogeneity index *HI*, b) Similarity index $SI_0$ with respect to proportions of the first observation, c) Similarity index $SI_\Delta$ with respect to proportions of the preceding observation, d) Kullback-Leibler heterogeneity *KL* e) Kullback-Leibler difference $KL_0$ with respect to proportions of the first observation, f) Kullback-Leibler difference $KL_\Delta$ with respect to proportions of the preceding observation. Confer text for definitions of these measures.

we conclude that measures based on information differences, due to their logarithmic dependence, weight larger differences in proportions stronger than small differences, whereas the ordinary measures exhibit a proportional weight.

In the same line, the time courses of measures of diversity and heterogeneity, respectively, with respect to active agents are depicted in Figs 7 and 8. Qualitatively, similar results as for the antibiotic classes are obtained. Once more, as already observed for the coefficient of variation, we see a damping effect of pooling the active agents into antibiotic classes. The variations of temporal changes of heterogeneity and related measures are larger for the active agents than for the more coarse grained antibiotic classes. No need to mention, the question of which stratification level should be prioritised is a matter of the concrete studies' objectives and the availability of adequate data. The usage of the more fine-grained level of active agents is advisable if records of prevalence of antibiotic resistance are available at the same fine-grained level.

Finally, we briefly report on the results obtained when the hospital's functional units are included as a second factor in addition to the antibiotic classes. Firstly, Fig 9 shows time courses of diversity measures stratified by the three functional units as previously defined. Secondly, heterogeneity *HI* and the similarity indexes $SI_0$ and $SI_\Delta$ are shown in Fig 10. Strikingly, diversities $D_a$ and *GS* as well as heterogeneity *HI* remain approximately constant in the course of time for functional unit 3 and varies only very slightly for unit 1. To the contrary, the diversities for functional unit 2 resemble the corresponding curves for the whole hospital as shown in Figs 5 and 6, i.e., they exhibit an increase in the course of time. Apparently, the functional units have different policies of antibiotic administration. This becomes even more obvious when throwing a glance

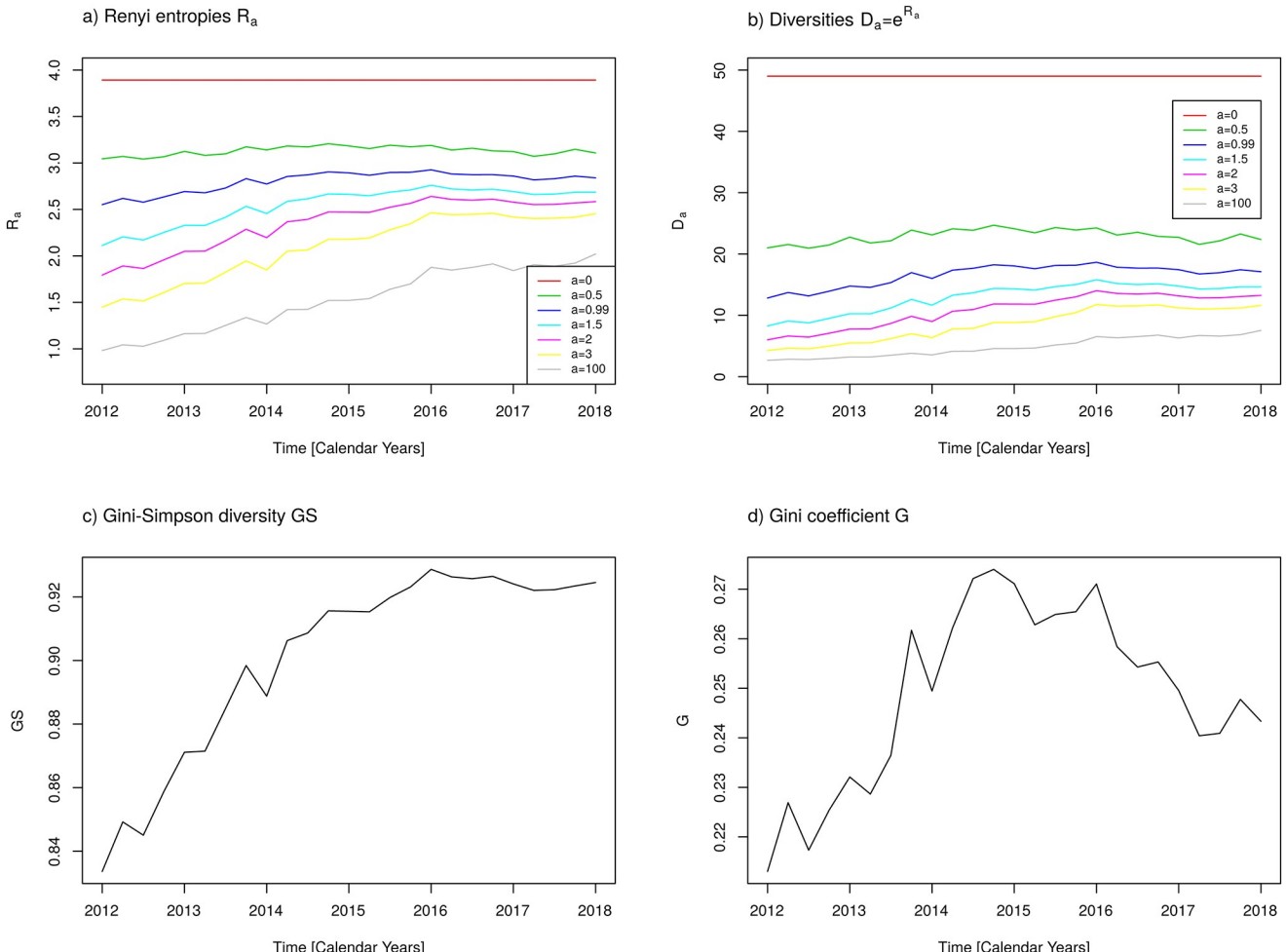

**Fig 7. Diversity measures with respect to active agents.** a) Renyi entropies $R_a$ for $a = 0, 0.5, 0.99, 1.5, 2, 3, 100$, b) Diversities $D_a$ for $a = 0, 0.5, 0.99, 1.5, 2, 3, 100$, c) Gini-Simpson index GS, d) Gini coefficient G.

onto the curve $SI_0(t)$ shown in Fig 10b. The administration in unit 3 essentially remains constant with respect to the first observed administration in 2012. The administrations in unit 1 and unit 2, in contrast, show a cumulative difference with respect to the first observation.

So far, we conclude that neither of the diversity or heterogeneity measures $D_a$, GS, G, HI, and KL are capable to catch suspected policies of clinical cycling without a concomitant assessment based on the newly introduced similarity indexes $SI_0$ and $SI_\Delta$ or, alternatively, $KL_0$ and $KL_\Delta$. A very weak long-term clinical cycling, as actually observed for the data under investigation, can leave diversity more or less invariant. To be specific, the constant non-vanishing slope of $SI_0(t)$, thus the constant $SI_\Delta(t) < 1$ reflects a long-term "cycling-like" change of antibiotic abundances. Assuming a full cycle in case of a rigorously applied cycling protocol, $SI_0(t)$ would also exhibit a full cycle in the course of time.

## Correlation of antibiotic administration and prevalence of antibiotic resistance

The most important and at the same time most challenging question, in the given context, is whether the clinical cycling of antibiotic administrations correlates or even causally relates to

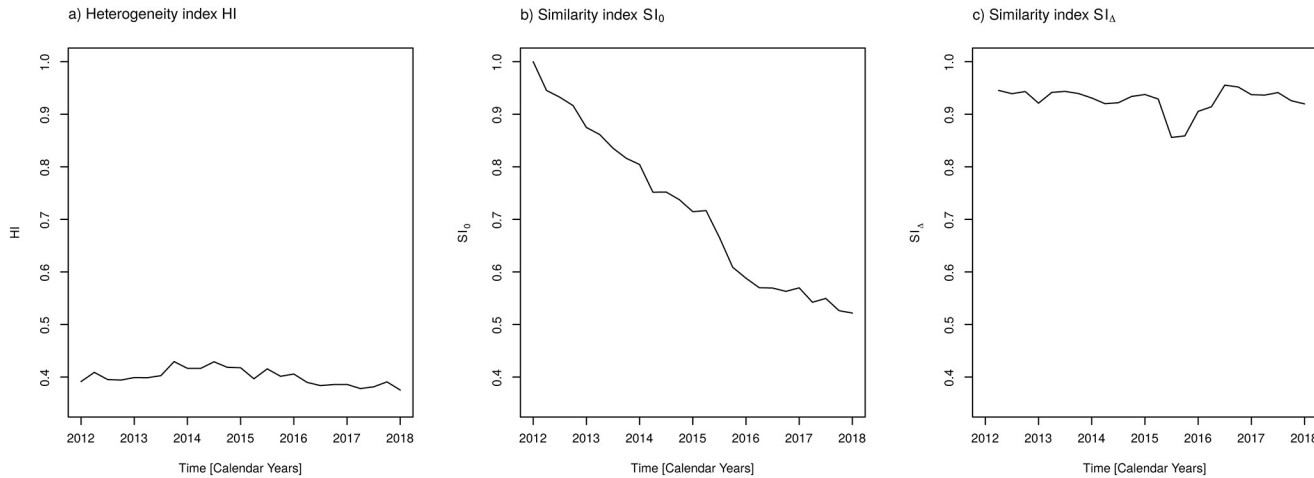

**Fig 8. Measures of heterogeneity and similarity with respect to active agents.** a) Heterogeneity index *HI*, b) Similarity index $SI_0$ with respect to proportions of the first observation, c) Similarity index $SI_\Delta$ with respect to proportions of the preceding observation. Confer text for the definitions of these measures.

the prevalence of antibiotic resistances. Only sufficient knowledge about existence and structure of such an association renders the design of administration policies that aim in minimising resistances meaningful. Unfortunately, recorded data on prevalence of antibiotic resistance are rare and often collected in a non-systematic way. Therefore, the following analysis should be viewed as paradigmatic rather than taking the results as credible.

Infections have been recorded on a yearly basis within intensive care units and medical/normal care units, however, not in a controlled and regular way as may be required by a controlled study design. Fig 11 shows the time courses of the number of registered cases per pathogen stratified by resistance. Resistance, hereby, has been dichotomised in a yes/no-variable although for some cases a more detailed information on the type of resistance (the corresponding antibiotic agent, multiresistance, etc.) is available. The time courses of infection frequencies suggest a rising prevalence. However, the awareness of the problem of antibiotic resistance and the adherence to diagnostic and therapeutic guidelines increased over time. We suspect that the rigorous diagnostic of pathogens is responsible for the added detection of more (resistant) pathogens. The proportions of infections with resistant infectious agents per type of pathogen is perhaps more reliable than the total number of infections. Fig 12 shows the time courses of these proportions and it no longer appears as drastic as before. It appears natural to apply the measures of heterogeneity and similarity introduced above to the proportions of resistant pathogens with respect to the total population of resistant pathogens.

Fig 13 depicts the time courses of *HI*, $SI_0$, and $SI_\Delta$ both for the proportions of antibiotic consumption and for the proportions of resistant pathogens in order to allow for a direct comparison. Heterogeneity hardly changes in the time's course both for antibiotic consumption and resistant pathogens (Fig 13, left panel). The slopes of heterogeneity *HI*(*t*) (denoted by mean (2.5%CI; 97.5%CI) in the following) resulting from linear regression are essentially zero (which is the null hypothesis of the linear model) with − 0.004 (− 0.012; 0.004) and $p = 0.300$ (antibiotics, unit 1), with 0.001 (− 0.003; 0.005) and $p = 0.705$ (pathogens, unit 1), with − 0.003 (− 0.021; 0.016) and $p = 0.629$ (antibiotics, unit 2), and with − 0.006 (− 0.019; 0.008) and $p = 0.300$ (pathogens, unit 2), respectively. However, heterogeneity of the population of resistant pathogens is slightly lower compared to antibiotic consumption. Thus, we have constant

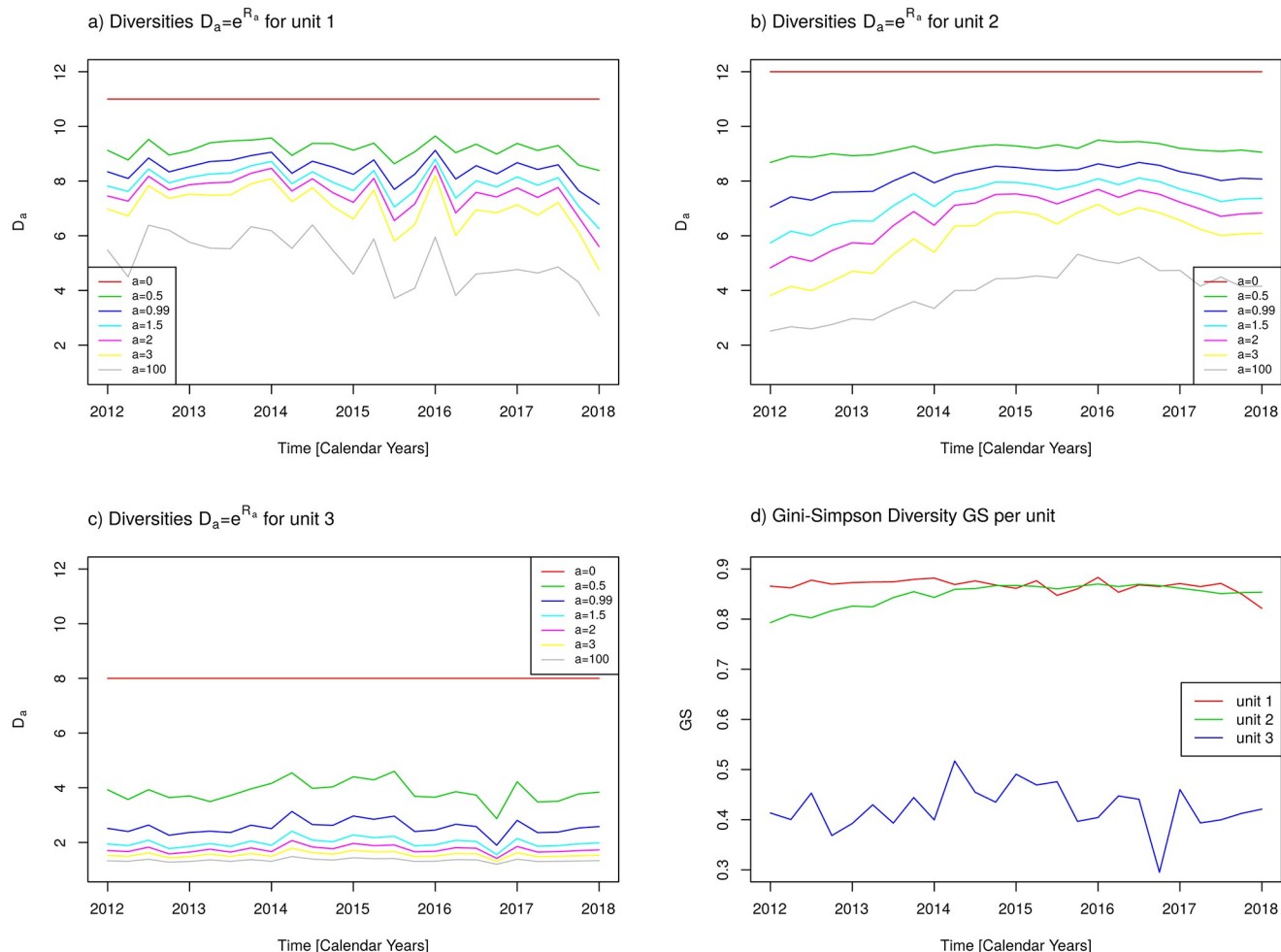

**Fig 9. Measures of diversity with respect to antibiotic classes stratified by functional units.** a) Diversities $D_a$ for unit 1 with $a = 0, 0.5, 0.99, 1.5, 2, 3, 100$, b) Diversities $D_a$ for unit 2 with $a = 0, 0.5, 0.99, 1.5, 2, 3, 100$, c) Diversities $D_a$ for unit 3 with $a = 0, 0.5, 0.99, 1.5, 2, 3, 100$, d) Gini-Simpson Index GS. See text for definitions.

heterogeneity but pathogens may, as observed for antibiotics, exhibit a "cycling-like" characteristic in form of exchanges of prevalences of pathogens which leave heterogeneity invariant.

The differential (quarter-by-quarter or year-by-year) changes of distributions with time averages $1 - SI_\Delta$ of 0.1 (0.08, 0.11) (antibiotics, unit 1), 0.15 (0.12, 0.18) (pathogens, unit 1), 0.053(0.05, 0.06) (antibiotics, unit 2), and 0.11(0.07, 0.14) (pathogens, unit 2), are slightly greater for the distribution of resistant pathogens compared to antibiotics consumption (Fig 13, right panel), however, both are constant in essence for both units with slopes (derived from a linear model) 0.0002 ($- 0.0078; 0.0081$) and $p = 0.968$ (antibiotics, unit 1), with $- 0.0091$ ($- 0.0460; 0.0279$) and $p = 0.492$ (pathogens, unit 1), with $- 0.0010$ ($- 0.0053; 0.0034$) and $p = 0.653$ (antibiotics, unit 2), and with 0.021 ($- 0.002; 0.045$) and $p = 0.0637$ (pathogens, unit 2). Therefore, we expect that $SI_0(t)$ exhibits a linear decline with constant slope approximately given by $SI_\Delta - 1$, as shown in the following.

Most strikingly, the time series of the accumulated similarity index $SI_0$ for the resistant pathogens strongly correlates with the corresponding time series of antibiotic consumption, in fact in both units. A Pearson product moment correlation analysis gives correlation

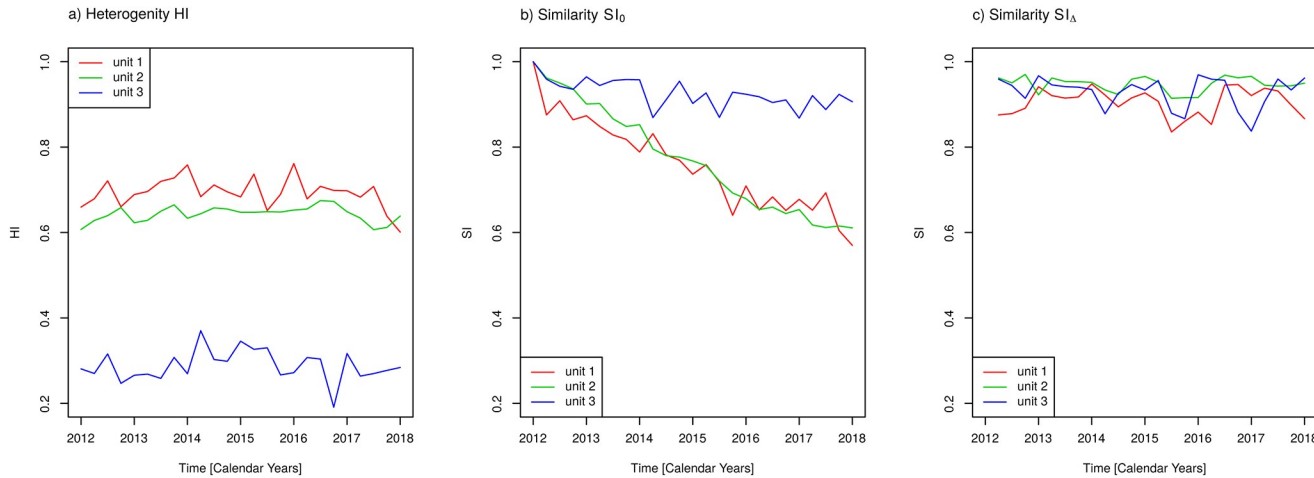

**Fig 10. Heterogeneity and similarities with respect to antibiotic classes stratified by functional units.** a) Heterogeneity $HI$. b) Similarity index $SI_0$ with respect to proportions of the first observation, c) Similarity index $SI_\Delta$ with respect to proportions of the preceding observation. Confer text for the definitions of these measures.

**Fig 11. Pathogen prevalence.** Sum of yearly registered number of cases of 9 observed pathogens stratified by resistance.

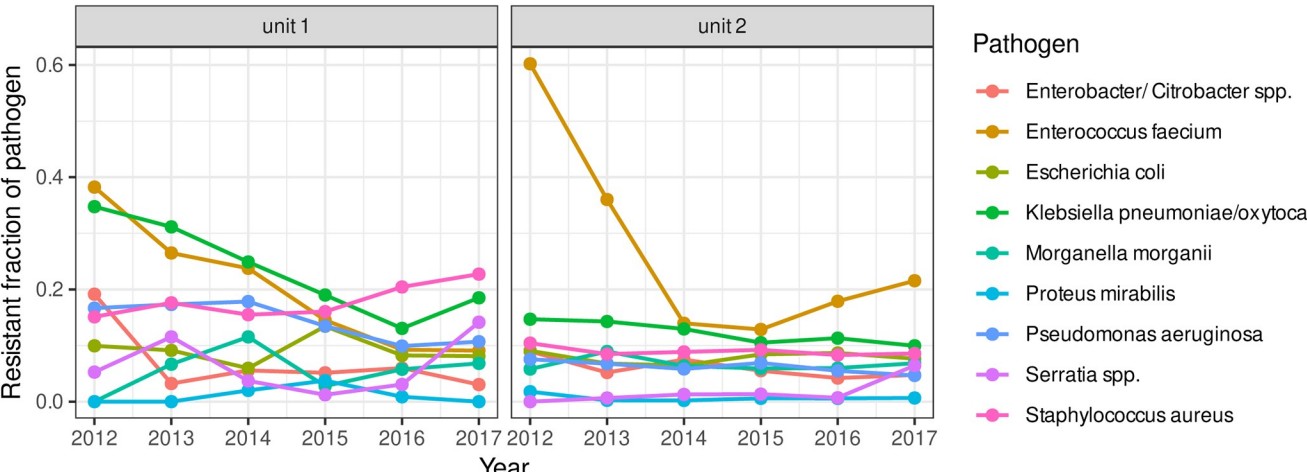

**Fig 12. Fraction of antibiotic-resistant germs.** Time courses of the fractions of resistance per pathogen plotted separately for units 1 (intensive care) and unit 2 (normal care unit).

coefficients 0.92 with $p = 0.009$ (unit 1) and 0.89 with $p = 0.02$ (unit 2). The slopes significantly differ from zero with the concrete values $- 0.055$ $(- 0.062; - 0.047)$ per year (antibiotics, unit 1, $p < 10^{-3}$), $- 0.049$ $(- 0.078; - 0.021)$ per year (pathogens, unit 1, $p = 0.009$), $- 0.067$ $(- 0.072; - 0.063)$ per year (antibiotics, unit 2, $p < 10^{-3}$), and $- 0.044$ $(- 0.088; - 7.3e - 04)$ per year (pathogens, unit 2, $p = 0.048$), respectively. It is appealing to speculate whether these coinciding changes are a result of correlations or even causal relations. For the time being, this speculation has to be treated with caution. However, this analysis gives directions to a proper controlled observational or experimental study design.

A further observation underpins our speculation. The total percentage of resistant pathogens reduces significantly from roughly 20% to 10% in functional unit 1. A linear regression gives a slope of $- 0.017$ $(- 0.025; - 0.008)$ per year for the proportion (significantly different from zero with $p = 0.005$). In functional unit 2 the proportion of resistant pathogens reduces non-significantly by $- 0.007$ $(- 0.015; 7.1e - 05)$ per year, however, at the edge of significance ($p = 0.051$). An additional support for our hypothesis is given by explicitly calculating the

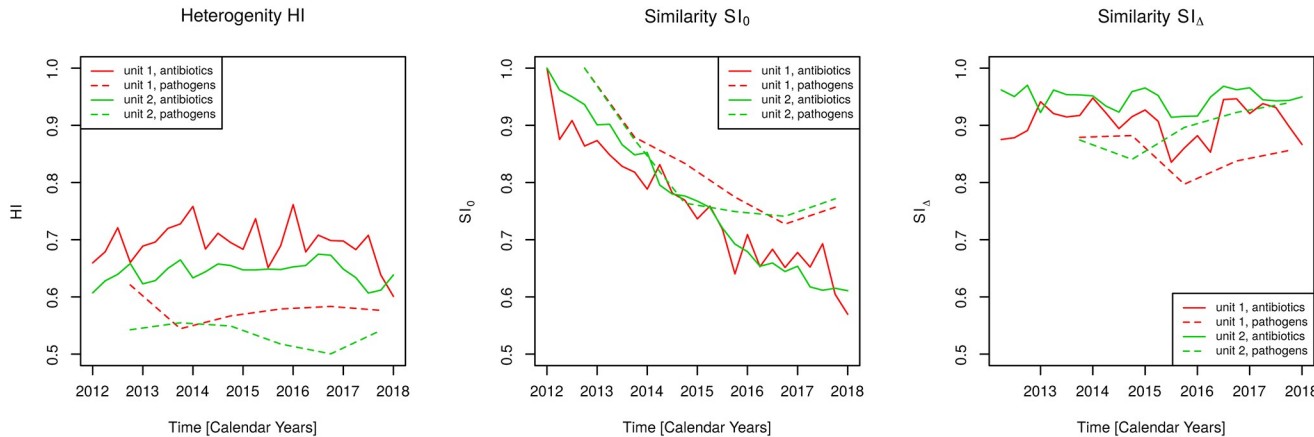

**Fig 13. Association of heterogeneities of antibiotic consumption and pathogen prevalence.** Time courses of heterogeneity, $HI$ and similarity indexes, $SI_0$ and $SI_\Delta$ for antibiotic consumption with respect to antibiotic classes and prevalence of resistant pathogens stratified by the hospital's units 1 (intensive care) and 2 (medical/normal care).

correlation coefficients between $SI_0$ and the approximately linear decline of the ratio of resistant pathogens: 0.92 ($p$ = 0.009) for unit 1 and 0.86 ($p$ = 0.03) for unit 2.

To conclude, although a genuine control strategy in the common sense of antimicrobial cycling or mixing was not applied, we observe that a rather long-term temporal change (clinical cycling) of consumption of different antibiotics ($SI_0$ applied to antibiotics consumption) correlates with a change of prevalence of antibiotic-resistant bacteria ($SI_0$ applied to prevalence of resistant pathogens). This correlation is expressed by means of almost equal slopes as well as a corresponding large correlation coefficient, where the slopes are significantly different from zero and approximately equal, of the two similarity indexes $SI_0$ for antibiotics and pathogens, respectively. Whether these temporal changes in antibiotic consumption have a direct causal impact is still speculative but gains additional evidence through the observed reduction of prevalent resistant bacteria. Controlled studies that allow comparisons with more or less static "control strategies" and other types of switching behaviours (including mixing) are needed to draw reliable inferences. It is the main intention of this work to supply an appropriate mathematical framework for such studies.

## Discussion

Applications of measures of heterogeneity and diversity are rare and unsatisfactory in the context of assessing antibiotic resistance. This is somewhat surprising since antibiotic administration policies that rely on cycling or mixing strategies in order to reduce antibiotic resistances have been promoted for quite some time [2, 4, 6, 7, 13] (for a counter example see [5]). Cycling strategies, this is our claim, are best characterised by means of differential measures of heterogeneity and diversity, respectively. This approach can, in principle, be extended to capture mixing by introducing a spatial stratification of the diversity measures. Although some attempts to tackle antibiotic resistance by means of heterogeneity analyses exist [2, 4, 13], a satisfactory mathematical framework is due.

We adopted diversity measures known in other fields of research [9, 11] and adapted them to the needs within the scope of analysing antibiotic resistance. It is natural to seek for dependencies between the heterogeneity of consumption of antibiotics and the heterogeneity of the pattern of prevalence of antibiotic-resistant pathogens.

In order to provide a flexible methodological basis for the analysis of antibiotic resistance, we introduced and discussed a simple measure of heterogeneity as well as a general family of diversity measures, i.e., the so called family of Renyi diversities and derivatives thereof. It should be noted that the notions of "heterogeneity" and "diversity" do not refer to conceptually different measures, they merely reflect their emergence in different fields of application. As a novel aspect within the given context, we derived differential measures of similarity which are needed to capture temporal changes due to swapping proportions which leave moments like heterogeneity and diversity invariant.

For many real-world applications, the simple heterogeneity measure $HI$ and differential measures of similarity $SI_0$ and $SI_\Delta$ will suffice. However, showing simultaneously that the whole family of diversity measures leads to the same conclusions supplies additional evidence ("We can regard a sample more diverse if all of its Renyi diversities are higher than in another samples.", [10]). Moreover, the smoothing and non-linear weighting effect of higher order measures like Shannon entropy and derivatives (Kullback-Leibler heterogeneity, etc.) might become important for damping spurious fluctuations by weighting larger deviations. A solid reason for the choice of entropies is the straightforward application of a maximum entropy method. Maximum entropy proved as the method of choice when it comes to learn dynamics of biological systems (e.g. [16], see also [11]). With the aid of such an optimisation tool we

expect that an optimal cycling and arguably also an optimal mixing schedule can be learned from the observed correlation patterns between antibiotics consumption and prevalences of antibiotic-resistance.

The presented inclusion of covariates and factors like clinical units and groupings of active agents has exemplary character. The concrete choice of covariates depends on their availability and, most important, on the specific questions that are raised. In the case of mixing with two (or more) subpopulations of patients that receive different drugs in a temporarily alternating way, it might be better to stratify for these subpopulations instead of functional units, unless these strata coincide. Furthermore, since transmission occurs at the microlevel, it would certainly be of advantage to include individual-level administration data instead of aggregated dispensing data. Such microlevel data have not been available for our elementary methodological approach. However, our analytical framework is flexible enough to account for such peculiarities. In addition, we point to the possibility to expand measures of heterogeneity and similarity to be applicable to joint probabilities of antibiotic consumption and resistance. This is beyond the scope of the present work, however, we paved the way for doing so.

It deserves to be mentioned, that some authors approached the problem by means of extended SIR-like epidemiological models [4, 17]. From a theoretical point of view, these models have benchmark character. However, the validation of these models necessitates recording of data on antibiotics consumption and pathogen load on an individual basis which is not feasible for most hospitals. As opposed to this, so called composite indices as "summary measures of the net impact of antibiotic resistance on empiric therapy" [18] are much more coarse-grained epidemiological measures based on the cumulative antibiogram [19], which reside on a higher population level. Our approach is compatible to both sides and bridges the gap.

In addition, our method complements time series analyses (e.g. [8]) that pointed to thresholds in associations between population antibiotic use and prevalence of resistant pathogens. Within the scope of the time series analyses as discussed by López-Lozano et al. [8], the correlation of the time series of the prevalence of a specific pathogen with the time series of the amount of corresponding administered antibiotics is calculated. Commonly, such a correlation of two time series is given by a mutual entropy. Thus, our approach is a generalisation in that it treats diversities of both antibiotic consumption as well as pathogen prevalence and correlates these diversities.

The time series analyses [8] supplied evidence that prevalence rates increase in a nonlinear fashion when exceeding a prevalence threshold after a sufficiently long duration of administration of certain antibiotics. The existence of such a threshold indicates that a switch to an alternative antibiotic agent is due. Our approach goes a step further by including the dynamics of switching in the analysis to allow, eventually, for an optimised (temporal and/or spatial) switching strategy. Results of stochastic simulations of microbial populations subjected to a periodic presence of antimicrobials [20] boost our confidence.

Moreover, due to its intermediate complexity it is able to serve as a performative boundary object [21], thence constituting a clinically relevant basis for a modelling for policy [22]. This holds all the more if implemented on a boundary infrastructure [23] as, for example, the modelling platform MAGPIE [24] that enables experts with different expertises to dock on. In other words, the proposed method has the potentiality to be translated to the point of decision making as a monitoring system.

## Conclusion

To conclude, the presented analysis has paradigm character. We focused on setting up a methodological framework because the available data do not allow to assess cycling or mixing

strategies in a controlled way. To be exact, a control strategy in a genuinely defined sense of cycling or mixing appears not to have been applied. In other words, we have a purely observational situation exhibiting a weak long-term "clinical cycling" but without ample background information particularly on individual-level administration. However, the performed applications of the suggested analytic methods to records of antibiotic consumption and prevalence of antibiotic-resistant bacteria definitely go beyond mere illustrations. That is to say, the results allow to raise hypotheses or at least to formulate conjectures. Specifically, we observe a strong positive correlation of time courses of similarity with respect to the initial observation of antibiotic consumption and prevalence of antibiotic-resistant pathogens. In addition, clinical cycling correlates with a decreasing ratio of resistant pathogens. These correlations have to be confirmed in an experimental/interventional study. We are convinced that the derived mathematical framework provides a sound basis to substantially improve the determination of a viable roll back administration policy to defeat antibiotic resistance.

## Supporting information

**S1 Data.**
(GZ)

**S1 File.**
(PDF)

## Author Contributions

**Conceptualization:** Hans H. Diebner, Katja de With.

**Data curation:** Anna Kather, Katja de With.

**Formal analysis:** Hans H. Diebner.

**Methodology:** Hans H. Diebner.

**Project administration:** Anna Kather, Katja de With.

**Supervision:** Ingo Roeder.

**Validation:** Ingo Roeder, Katja de With.

**Visualization:** Hans H. Diebner.

**Writing – original draft:** Hans H. Diebner.

**Writing – review & editing:** Hans H. Diebner, Anna Kather, Ingo Roeder, Katja de With.

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
