## [Decision Letter · Decision Letter 0]

27 Feb 2020

PONE-D-19-25350

Mathematical basis for the assessment of antibiotic resistance and administrative counter-strategies

PLOS ONE

Dear Dr. Diebner,

Thank you for submitting your manuscript to PLOS ONE. After careful consideration, we feel that it has merit but does not fully meet PLOS ONE’s publication criteria as it currently stands. Therefore, we invite you to submit a revised version of the manuscript that addresses the points raised during the review process.

As you will see in her/his detailed comments below, Reviewer 2 raises very important concerns about your study and how your conclusions are supported by your results and analysis (shared with Reviewer 1). In order for your study to be recommended for publication in PLOS One, all these points must be addressed in deep detail. 

We would appreciate receiving your revised manuscript by Apr 12 2020 11:59PM. To enhance the reproducibility of your results, we recommend that if applicable you deposit your laboratory protocols in protocols.io, where a protocol can be assigned its own identifier (DOI) such that it can be cited independently in the future. For instructions see: http://journals.plos.org/plosone/s/submission-guidelines#loc-laboratory-protocols

We look forward to receiving your revised manuscript.

Kind regards,

Ricardo Martinez-Garcia

Academic Editor

PLOS ONE

Journal Requirements:

Journal Requirements:

Reviewers' comments:

Reviewer's Responses to Questions

**Comments to the Author**

1. Is the manuscript technically sound, and do the data support the conclusions?

Reviewer #1: Partly

Reviewer #2: No

2. Has the statistical analysis been performed appropriately and rigorously? 

Reviewer #1: Yes

Reviewer #2: No

3. Have the authors made all data underlying the findings in their manuscript fully available?

Reviewer #1: Yes

Reviewer #2: Yes

4. Is the manuscript presented in an intelligible fashion and written in standard English?

Reviewer #1: Yes

Reviewer #2: Yes

5. Review Comments to the Author

Reviewer #1: Authors perform a rigorous mathematical analysis on the observational data, use multiple measures to asses the diversity and the heterogeneity of antibiotic consumption, and connect their output to the resistance prevalence observed in the dataset.

Although I think the approach is very valuable, I think there are major points that need to be modified before the publication of this manuscript. My comments are below.

Major Points

1) I don't understand how measures of diversity or heterogeneity are connected to mixing (spatial heterogeneity) or cycling (temporal heterogeneity) strategies here. Mixing refers to giving different antibiotics to the different members of the cohort, but that can also end up having a constant DDD over time (say you keep 50% of the population on one antibiotic and the other 50% on the other, but the point is to alternate between specific individuals.). Similarly, cycling - the temporal approach - should have way more variation over time, such as excluding certain antibiotics completely from the hospital or the cohort. So I cannot relate the antibiotic consumption observed in this dataset with the common strategies used.

2) The authors' most important conclusion is only expressed in one sentence, and not fully clear (lines 362-364). Do they mean almost a no slope by saying that " means of parallel slopes "? Is this really a good way to back their argument up? Given that this is the message of the paper, I really expect more elaboration on this point.

3) In general, I understand why authors use so many different measures to asses diversity or heterogeneity, and while reading the manuscript, it was obvious that this is the way they explored the whole research question. But is it necessary to include all these measures in the main text? There is a lengthy section about the explanation and use of so many different measures, but the conclusion and the connection to resistance prevalence is relatively really short, which is supposed to be the main point. I think the manuscript needs a bit of structural revision in this regard.

Minor Points

1) Figures 4-10 : please also use dots for the data points, like you did for figures 1-3.

2) line 14 : these strategies are not really "recent" anymore.

3) lines 19-21 : please refer to the key publications that includes the "evidence" you have mentioned.

Reviewer #2: Diebner and collaborators present data about antibiotic consumption and pathogen resistance in three administrative units of a hospital over several years. They aim at using these data to test hypotheses related to the efficiency of heterogeneous antibiotic treatments strategies to mitigate the emergence of resistance in hospitals. For this they suggest the use of various measures of heterogeneity (that are classically used in theoretical ecology) in the context of antibiotic treatment and resistance.

I am not convinced of the interest of this framework for the analysis of heterogeneous antibiotic treatment strategies, for 3 reasons:

- The measures that go beyond the coefficient of variation are interesting when strictly more than 2 antibiotics are in use. In the "antibiotic mixing" literature, to my knowledge the vast majority of the literature considers the case of 2 antibiotics (the question being whether treating half of the patients with AB1 and half with AB2 is better than treating everybody with AB1, or better than cycling, see further below).

- These measures are used in ecology to deal with very heterogeneous data, eg when abundances are very different between different species. This is useful (in ecology) because a species may be several orders of magnitude less abundant than another but still be essential for ecosystem function. But in the case of antibiotic treatments, the point of DDD is that different antibiotics can be compared between them. If the DDD of 2 antibiotics differ by orders of magnitude, it means that the one with the low DDD is virtually absent and has no chance of contributing to the establishment of resistance (or to infection clearance).

- In much of the literature (including the one cited by the authors) on heterogeneous antibiotic treatments, a big question is "mixing vs cycling", but I do not see how the measures suggested by the authors can be useful to study cycling. The authors simply do not ever mention cycling after the introduction. It gives me the impression that the papers they cite do not really match the question they want to study.

Most importantly, I am also not convinced that the presented data support the claimed result ("a reduction of prevalence of antibiotic-resistant germs correlates with a change of heterogeneity of antibiotics consumption"), for several reasons:

- Only time series are analysed. In time series, the different time points are not independent from each other, and it is thus not appropriate to decide whether the slope is significantly different from 0 using a linear regression or a correlation test (by the way the authors do not even precise which statistical test they use).

- Regarding panels 1 and 3 of fig 13 (HI and SIdelta), both the curves for heterogeneity / similarity of antibotic use and pathogen resistances are flat. The authors conclude that the curves for treatment and resistance are similar and thus that there methodology permits to establish that heterogeneous treatments cause heterogeneous resistance. There is absolutely no statistical support (nor proper testing) for this finding. And a more parsimonious explanation for the fact that both curves are flat could be that HI and SIdelta do not capture any property of the data and are always constant. Actually on all figures of the papers, all the temporal curves of HI and SIdelta are indeed flat. I think that questions the interest of these measures.

- Regarding pannel 2, the observation (that the slopes of SI0 are the similar for antibiotic use and pathogen resistance) is not uninteresting, but these are only two independent data points (two hospital units) and it is thus impossible to have any statistical support. The authors mention that before pooling all of them into 3 units, they have a larger number of administrative units. Maybe they could test their hypothesis at the level of these smaller units? I do not agree that "a grouping of these small subunits into functional units [is] sufficient" if they want to test this hypothesis.

- I think there is an important difference between the question presented in the introduction and in the literature (do mixing startegies reduce antibiotic resistance?) and the question mainly adressed in the results (do heterogeneity of treatment correlate with heterogeneity of resistance?). It is only in the last paragraph of the results that the authors attempt to adress the former.

So in the current state, I do not see how the methods and the data presented in this article permit to adress what the authors present as the main question, and can not recommend the publication of this article as long as this is not clarified. But I am sure that there is potential interest in the data presented by the authors (time series of antibiotic treatments and pathogen resistance in different units of a hospital).

Below are more detailed comments about the manuscript, in three parts (A: abstract, B: introduction and methods, C: results).

A) I did not find the abstract understandable before reading the rest of the article:

"Temporal changes of the proportional abundances of different antibiotics (e.g. mixing or cycling)": unlike cycling, mixing does not implies temporal changes but spatial heterogeneity

"Although such a mixing strategy appears to be plausible": at this point mixing is not defined, and the previous sentence seems to rather relate to cycling than mixing ("temporal changes").

"We adopt diverse measures of heterogeneity and diversity": the authors should at least precise the variable whose they are trying to quantify heterogeneity of diversity

"We show that a reduction of prevalence of antibiotic-resistant germs correlates with a change of heterogeneity of antibiotics consumption" -> I would suggest to precise the direction of the change in the abstract

"we introduce a scheme based on linear regression for the assessment of associations between changes of heterogeneities on the antibiotics and the pathogen side" and "we show that a reduction of prevalence of antibiotic-resistant germs correlates with a change of heterogeneity of antibiotics consumption" -> I do not see what is new in the method (what does it mean to "introduce a linear regression scheme"? Linear regression is not new, and here is not really adapted for time-series data as explained above). All the authors do is plotting two curves next to each other and say they look similar (fig 13), without proper statistical analysis.

B) The introduction and methods should be clarified, and need proper references to the literature:

L16 "whereas other strategies refer to a scheduled change of the dominantly used class of antibiotics": in the literature (including the one cited by the authors), antibiotic mixing does not involve scheduled change of antibiotic over time, but treatment of different subgroups of patients with different antibiotics.

L17 "a fraction of patients": I think the authors should introduce the context better, to explain that these strategies are defined at the scale of a group of patients (for example in a hospital). This is important because with the definition provided by the authors, cycling could be wrongly understood as applied to individual patients.

L19 "Although there is some evidence that": citations needed. Overall there is a huge lack of references to the literature in the article.

L23-25 I find the sentence very unclear. What does it mean to "quantify the heterogeneity of [...] time courses of prevalence of antibiotic-resistant pathogens" ?

L57 is "DDDi" referring to the DDD or the "consumption density DDD"? If the former, why mentioning "consumption density DDD"?

L56-57: At this point, it is not clear what is "the antibiotic group i". The previous paragraph explains that the hospital departments are clustered into 3 administrative units, and that the antibiotics are pooled in 12 antibiotic classes. To which kind of group this sentence refers to (are the authors computing antibiotic consumption for a class of antibiotic or for a unit of the hospital)? It only becomes clear much later in the article.

L60: Is the coefficient of variation computed for the vector of DDDi(t) for all i? Maybe clarify the formula?

The paragraph "Coefficient of variation" must include references. It raises an interesting but rather simple point, and could be shortened.

The paragraph "Heterogeneity and entropy" is mostly paraphrasing the literature without proper citations. The authors do not explain how these theoretical ecology measures will be applied to quantify antibiotic resistance and antibiotic uses. I understand that they want to stay general because they will apply the same measures to different data, but everything would be more clear if the "antibiotic groups" had been properly defined (see my previous comment)

L90, The definition of the "proportions" are not clear at this point. Do a1+a2+a3+...+an=1, or do ai+bi=1 for all i? Said otherwise, is "ai" the proportion of individuals of population a that belong to species i, or is it the proportion of individuals of species i that belong to environment a ? It only becomes clear much later in the text.

C) The data presented in the results (antibiotic consumption and resistance in a hospital) have some potential interest, but at this stage the authors do not show anything convincing.

Figure 1 : If I correctly understand, the fact that fig 1a and fig 1b look very similar suggest that the sum of DDD for all the antibiotic groups is almost constant over time. This should be made visible on Fig 1c by making the y-axis start at 0, otherwise a quick look at this panel conveys an inaccurate message.

L167-174 I do not see how what it presented in this paragraph is a result. The authors are mostly saying that the coefficient of variation is a good measure of variation.

L 192-293 "Please note, the different scales of the y-axes reflect the different total amounts of consumption within each unit due to their different sizes": if this really comes from difference in size of the units (number of patients), Isn't it a strong argument for normalising DDD per patient days? I do not understand why the authors describe this normalisation in the methods but do not use it later in the results while it seems needed and appropriate.

L305 "The most important and at the same time most challenging question, in the given context, is whether the mixing behaviour of antibiotic administrations correlates or even causally relates to the prevalence of antibiotic resistances.": the authors do not show that their methods are useful for such a test. Do they really bring something compared to the coefficient of variation?

Figure 12 mentions "intensive care" and "normal care" unit, while everywhere else in the article the units are labeled "unit 1", "unit 2" and "unit 3"

I do not understand the sentence "We conclude that mixing of antibiotic consumption correlates with the prevalence of antibiotic-resistant bacteria by means of parallel slopes of similarity indexes SI0"

Dataset: the authors made the effort of making the full dataset and analysis programs available. Just a minor suggestion: the variable names in R data sheets and the text in the R markdown file are currently in german, I think they would be more useful if translated to english.

6. PLOS authors have the option to publish the peer review history of their article (what does this mean?). If published, this will include your full peer review and any attached files.

Reviewer #1: No

Reviewer #2: No

---

## [Author Response · Author response to Decision Letter 0]

18 Jun 2020

Response to Reviewer #1:

Major Points

1) I don't understand how measures of diversity or heterogeneity are connected to mixing (spatial heterogeneity) or cycling (temporal heterogeneity) strategies here. Mixing refers to giving different antibiotics to the different members of the cohort, but that can also end up having a constant DDD over time (say you keep 50% of the population on one antibiotic and the other 50% on the other, but the point is to alternate between specific individuals.). Similarly, cycling - the temporal approach - should have way more variation over time, such as excluding certain antibiotics completely from the hospital or the cohort. So I cannot relate the antibiotic consumption observed in this dataset with the common strategies used.

Answer: Yes, it is true that cycling or mixing can end up in having the same diversity. This is one of the major aspects of our discussion and lead us to the introduction of a differential measure SI_0. In the revised version, we additionally derived Eq 10 which is appropriate to capture mixing. In fact, the discussion of SI_0 is the most crucial point of our analysis and elaborated on that in the revised version.

A general remark with respect to the dataset seems to be due. Main purpose of our work is to derive a sound mathematical framework for the analysis of antibiotic control strategies. The application to the data available for us has to be understood mainly as illustration of how the mathematical methods are applied. We emphasise the fact, that a control in a genuinely defined way either of cycling or mixing, respectively, has obviously not been applied in the Hospital from which we draw data. However, a long-term change of antibiotic administration can be observed which we call „clinical cycling“ following Abel Zur Wiesch et al. 2014. In the light of this rather weak temporal changing behaviour we feel inclined to formulate our (secondary!) finding with respect to the concrete observation in a positive way: Despite the relatively weak differential change of diversity in antibiotic consumption, we observe a positive correlation with the corresponding differential change of prevalent resistant pathogens. We strengthend this point in the revised version. 

This point is strongly related to the following objection of Reviewer #1 

2) The authors' most important conclusion is only expressed in one sentence, and not fully clear (lines 362-364). Do they mean almost a no slope by saying that " means of parallel slopes "? Is this really a good way to back their argument up? Given that this is the message of the paper, I really expect more elaboration on this point.

Answer: Throughout the revised manuscript, we clarified this point. In addition, with respect to the paragraph mentioned by Reviewer #1 (lines 362-364 of the first submission), we explicitly added/revised:

„To conclude, although a genuine control strategy in the common sense of antimicrobial cycling or mixing was not applied, we observe that a rather long-term temporal change (clinical cycling) of consumption of different antibiotics ($SI_0$ applied to antibiotics consumption) correlates with a change of prevalence of antibiotic-resistant bacteria ($SI_0$ applied to prevalence of resistant pathogens). This correlation is expressed by means of almost equal slopes as well as a corresponding large correlation coefficient, where the slopes are significantly different from zero and approximately equal, of the two similarity indexes $SI_0$ for antibiotics and pathogens, respectively. Whether these temporal changes in antibiotic consumption have a direct causal impact is still speculative but gains additional evidence through the observed reduction of prevalent resistant bacteria. Controlled studies that allow comparisons with more or less static ``control strategies'' and other types of switching behaviours (including mixing) are needed to draw reliable inferences. It is the main intention of this work to supply an appropriate mathematical framework for such studies.“

Please also note that we strengthened our arguments in the Results section where we discussed the corresponding findings of how SI_0 behaves.

3) In general, I understand why authors use so many different measures to asses diversity or heterogeneity, and while reading the manuscript, it was obvious that this is the way they explored the whole research question. But is it necessary to include all these measures in the main text? There is a lengthy section about the explanation and use of so many different measures, but the conclusion and the connection to resistance prevalence is relatively really short, which is supposed to be the main point. I think the manuscript needs a bit of structural revision in this regard.

Answer: It is indeed important for us to introduce and discuss the whole family of diversity measures within the main text and compare it with the antibiotic homogeneity index previously used by other authors. Although all members of the family share common features, the concrete choice nevertheless depends on the concrete structure of studies and related research questions to which the method will be applied. The discussion of features of the proposed measures and their comparisons are indeed crucial points of our manuscript. 

Minor Points

1) Figures 4-10 : please also use dots for the data points, like you did for figures 1-3.

Answer: We would like to stick with the rule to only represent actual data points and summary statistics/moments like mean, median, variance etc. by dots but not nonlinear functions like entropies etc. 

2) line 14 : these strategies are not really "recent" anymore.

Answer: Yes, we agree and changed it accordingly. 

3) lines 19-21 : please refer to the key publications that includes the "evidence" you have mentioned.

Answer: We added references that show the mentioned evidence including

 Sandiumenge et al. 2006

 Bennett et al. 2007

 AbelZurWiesch et al. 2014

 Lopez-Lozano et al. 2019

 Davey et al. 2013

and also one reference for a counter-example:

 Karam et al. 2016

Response to Reviewer #2:

Reviewer #2: Diebner and collaborators present data about antibiotic consumption and pathogen resistance in three administrative units of a hospital over several years. They aim at using these data to test hypotheses related to the efficiency of heterogeneous antibiotic treatments strategies to mitigate the emergence of resistance in hospitals. For this they suggest the use of various measures of heterogeneity (that are classically used in theoretical ecology) in the context of antibiotic treatment and resistance.

General answer that applies to several objections raised by Reviewer #2:

Main purpose of our work is to derive a sound mathematical framework for the analysis of antibiotic control strategies (cf. the title „Mathematical basis for the assessment ...“ ). The application to data available to us has to be understood mainly as illustration of how the mathematical methods are applied. We emphasise the fact, that a control in a genuinely defined way either of cycling or mixing, respectively, has obviously not been applied in the Hospital from which we draw data. However, a long-term change of antibiotic administration can be observed which we call „clinical cycling“ following Abel Zur Wiesch et al. 2014. In the light of this rather weak temporal changing behaviour we feel inclined to formulate our finding with respect to the concrete observation in a positive way: Despite the relatively weak change of diversity in antibiotic consumption, we observe a positive correlation with the decline of prevalent resistant pathogens. Throughout the revised manuscript, wherever appropriate, we more strictly pointed out our main intention to present a math framework rather than presenting the results of a clinical study. In addition, we strengthend the provisional and somewhat speculative results with respect to data analysis. Even in the virtual absence of „heterogeneous antibiotic treatments“ there is a weak but noticeable impact of the increase of diversity and, more important, of the changes with respect to the initial observation time (captured by SI_0) on the reduction of antibiotic resistance. 

We clarified and strengthened this point throughout the revised manuscript wherever appropriate. Also confer the following answers.

I am not convinced of the interest of this framework for the analysis of heterogeneous antibiotic treatment strategies, for 3 reasons:

- The measures that go beyond the coefficient of variation are interesting when strictly more than 2 antibiotics are in use. In the "antibiotic mixing" literature, to my knowledge the vast majority of the literature considers the case of 2 antibiotics (the question being whether treating half of the patients with AB1 and half with AB2 is better than treating everybody with AB1, or better than cycling, see further below).

Answer:

Cycling and mixing in their extreme forms are rarely met, which is why Abel zur Wiesch et al. 2014 introduced the notion of „clinical cycling“ for the more frequently met adjusted forms. In a real clinical situation, in the absence of a strict mixing plan (with only 2 AB involved), usually more than 2 AB are involved. 

Surely, in the case of a given strict mixing with only 2 AB and a single given type of infection (pathogen), one would switch to prospective trials (randomized clinical trials (RCTs), controlled clinical trials (CCT), controlled before-after, cross-over) based on well-defined rotation/switch protocols, thereby, however, excluding continuous real-world processes. Alternatively, one can perform a time series analysis as discussed in Lopez-Lozano et al. 2019. Here the prevalence time series of the resistant pathogen is correlated with the time series of the amount of administered AB. Usually, in such a correlation analysis of two time series, a mutal entropy is used to quantify the correlation. In fact, we see in our approach a generalisation to more than 2 involved AB but this does not exclude the application to only 2 AB.

We agree that we used the notions of cycling and mixing in a somewhat vague way. We added clarifying explanations and, most important, we added Eq. 10 in the revised version which explicitly accounts for mixing in the strict sense. Even in the case of only 2 AB these measure results in an interpretable output, although in such a case one would prefer to switch to the time series correlation analysis of 2 time series, as mentioned above. 

- These measures are used in ecology to deal with very heterogeneous data, eg when abundances are very different between different species. This is useful (in ecology) because a species may be several orders of magnitude less abundant than another but still be essential for ecosystem function. But in the case of antibiotic treatments, the point of DDD is that different antibiotics can be compared between them. If the DDD of 2 antibiotics differ by orders of magnitude, it means that the one with the low DDD is virtually absent and has no chance of contributing to the establishment of resistance (or to infection clearance).

We have problems in fully understanding this objection. A typical application in ecology is the impact of crop diversity on the vulnerability of the system to pests. One may hypothesise, that a monoculture (a low diversity) may lead to high vulnerability. This hypothesis is driven by real observations. This vulnerability might be reduced either by implementing a diverse crop culture or by annually switching the type of crop (including spatial switching=mixing) or a combination of that. We hope that this gives a vivid illustration that there are indeed many analogues between ecosystem and antibiotic stewardship. 

To quantify the impact of diversity as well as the impact of temporal and spatial changes that might leave overall diversity invariant, an appropriate measure is needed, which we presented (SI_0). We do not see why such measures should not be applicable to situations where a diverse (>2 species) system is compared to a 2-species-system including strategies of cycling and mixing. Furthermore, we do not see why this should not be translated into the area of antibiotic stewardship. By the way, with respect to DDD, even the reverse direction might be of interest, namely to introduce an abundance measure that renders crops comparable with each other by means of their contribution to vulnerability. This is of course much beyond the scope of our approach and we are lucky to already have a working definition of DDD.

More specific, given a fixed number of different species, diversity (thus heterogeneity in the ecological sense) reaches is maximum if all species are equally abundant, no matter of whether the whole ecosystem consists of only 2 or more different species. Whether or not diversity correlates positive or negative or not at all with a target parameter as, e.g., prevalence of resistant germs, has to be shown. This is the first of our objectives of presenting diversity measures and we are not quite sure what should be wrong with this approach. 

Relating the prevalence of resistant pathogens to diversity is what other authors approached earlier, e.g. Sandiumenge et al. 2006 or Pluess-Suard et al. 2013, however, using the antibiobtic heterogeneity index (HI in our notation) only. Two things should be mentioned in this context. Firstly, looking at the mere diversity may not suffice, but one should also treat the effect of changes of the amount of AB even if these changes leave diversity invariant. This is what we did when introducing differential measure SI_0. Whether such effects exist or not is still open, but we supply the proper method to test it in the future, and, importantly, we supplied initial evidence through our analysis. Secondly, we used other (better in certain contexts) measures in form of Renyi entropies and compared them to HI. HI fails to capture weak changes of diversity. If low DDD of a given AB really has no impact, than it should be reflected in the impact of a low diversity. 

- In much of the literature (including the one cited by the authors) on heterogeneous antibiotic treatments, a big question is "mixing vs cycling", but I do not see how the measures suggested by the authors can be useful to study cycling. The authors simply do not ever mention cycling after the introduction. It gives me the impression that the papers they cite do not really match the question they want to study.

We agree that it has not always been clearly formulated of whether „only“ diversity is addressed or whether changes in administration (cycling, mixing) are addressed that might in principle leave overall diversity invariant. We addressed this lack of clear formulation in the revised version of our manuscript. Particularly, we introduced Eq 10 in the revised version that explicitly captures mixing strategies. Most important, SI_0 captures cycling very well and we put more effort into explaining this fact in the revised version. 

Most importantly, I am also not convinced that the presented data support the claimed result ("a reduction of prevalence of antibiotic-resistant germs correlates with a change of heterogeneity of antibiotics consumption"), for several reasons:

- Only time series are analysed. In time series, the different time points are not independent from each other, and it is thus not appropriate to decide whether the slope is significantly different from 0 using a linear regression or a correlation test (by the way the authors do not even precise which statistical test they use).

Answer: Yes, time series are often autocorrelated, but also non-autocorrelated time series exist. More important beyond this general remark, two time series may be mutually correlated. Why is the use of time series analyses formulated as an objection or as a limitation („ONLY time series are analysed“)? It is the very essence and the strength of our processual approach to evaluate whether such a mutual information (correlation) of two time series (antibiotic consumption and pathogen prevalence) exists. For an exquisite example that points in this direction see Lopez-Lozano et al. 2019. 

We have quarterly data over several years and why should we not use this information on the observed time courses? Moreover, the processual approach is applicable to field-like real-world observational studies. If sequential observations at several observation time points are given, why should one not use this time series? The crucial point is to aggregate the time series of many abundances into a manageable and meaningful proxy time series: diversity and/or differential similarity.

Unfortunately, for the data available to us, we cannot attribute time series of specific antibiotic treatment to the emergence of a specific resistant pathogen as in Lopez-Lozano et al. What we have is the time series of diversity (HI or Renyi entropies) and (auto-) similarity measures (e.g. SI_0) and we relate these time series for antibiotics and pathogens with each other. We observe a correlation. Moreover, SI_0 is a measure which captures adjusted administration of AB including cycling even if diversity is unchanged. Unfortunately, for our data, we do not observe a full cycle but rather a long-term change which can be interpreted as the onset of cycling (weak clinical cycling). In our case, the adjusted administration appears in form of a linear change of SI_0 and we think it is legitimate to ask whether this change differs significantly from zero (by means of inference based on confidence). 

 As long as we do not have experimental (prospective) data but have to rely on an observational study only, we cannot conclude that this correlation is due to causality. The only thing we can do is raising a hypothesis, which is what we did. 

Perhaps, using a more complex multivariate statistical approach to time series would be the ultimate improvement. However, using diversity and similarity measures, as we did, as surrogate measures to capture essential aspects of the time series appears to be a legitimate approach with medium (i.e. comprehensible) complexity. 

In the revised version, we elaborated on our processual stance mostly within the introduction but also throughout the manuscript. In addition, we emphasised the relevance of SI_0 and we extended the according analysis.

- Regarding panels 1 and 3 of fig 13 (HI and SIdelta), both the curves for heterogeneity / similarity of antibotic use and pathogen resistances are flat. The authors conclude that the curves for treatment and resistance are similar and thus that there methodology permits to establish that heterogeneous treatments cause heterogeneous resistance. There is absolutely no statistical support (nor proper testing) for this finding. And a more parsimonious explanation for the fact that both curves are flat could be that HI and SIdelta do not capture any property of the data and are always constant. Actually on all figures of the papers, all the temporal curves of HI and SIdelta are indeed flat. I think that questions the interest of these measures.

Answer: HI (and likewise the Renyi entropy) measures per definition diversity. It remains constant for the observed data. This is, in our opinion, a relevant and most illuminating result. We do not understand why diversity should not capture a relevant property of the data. 

If the abundances of a low-abundant and a highly-abundant species are exactly swapped, diversity/heterogeneity does not change. This means, diversity/heterogeneity (particularly HI) does not capture the swap of species abundances. The very strength of our approach is, that a measure for the strength of cycling can indeed by derived: SI_0. 

So far, to the best of our knowledge, only HI has been used in the literature to describe antibiotic diversity which does not capture such a swap. For our data, HI is almost constant over time, although throwing a glance onto the time courses of the consumptions confirms that indeed the amount of essentially three antibiotics switched in the course of time.

Thus, we think that it is one of the strengthes of our work having shown that HI or any other pure diversity does not suffice to capture relevant changes in consumption. To capture changes in abundances – INCLUDING those which leave diversity invariant – we introduced SI_0 which detects whether a change took place or not.

With respect to the analysed data, there is a more or less constant shift of some of the antibiotics abundances, which is captured by a linear decline of similarity with respect to the first observation. SI_0. The slope is a measure of the intensity of these shift. SI_Delta is an approximation to this slope and the magnitude of 1-SI_Delta is an exquisite measure of the strength of cycling. It is weak and constant in our case, but this is a particular and relevant finding for this case. 

Thus, HI (i.e. diversity) might in concrete real observations, as is the case for our data, turn out to exhibit no change at all, although there have been changes in consumption. Cycling and mixing might leave heterogeneity/diversity invariant, therefore, we introduced a „differential“ measure to account for such changes. In a sense it is a nice coincidence that our data exhibits an almost constant diversity, although changes in administration happened, because this supplies additional evidence (in form of a proof-of-principle) for the need of our newly introduced „differential diversity measure“ SI_0. SI_Delta is in a certain sense redundant, since it captures the slope of SI_0 which is constant in case of linear SI_0. However, it is arguably a good index that quantifies the strength of cycling in a general sense, including adjusted forms of AB prescriptions.

- Regarding pannel 2, the observation (that the slopes of SI0 are the similar for antibiotic use and pathogen resistance) is not uninteresting, but these are only two independent data points (two hospital units) and it is thus impossible to have any statistical support. The authors mention that before pooling all of them into 3 units, they have a larger number of administrative units. Maybe they could test their hypothesis at the level of these smaller units? I do not agree that "a grouping of these small subunits into functional units [is] sufficient" if they want to test this hypothesis.

Answer: We completely agree, for a statistical support particularly of the impact of mixing, it is necessary to look at a more fine-grained level of administrative units. „Sufficient“ here refers to the sufficiency for the illustrative purpose with respect to the math framework. We added Eq 10 in the revised version that clarifies this point. However, we would like to keep the focus on the math framework instead of going into depth with the analysis of a dataset that has been recorded in the absence of a well-defined or documented administration strategy. Particularly, information on unit-specific prevalences is missing. We think that making the math framework public even without the analysis of an adequate controlled study has priority. 

Moreover, our quantification will become most important when being compared to the outcome of other hospitals. Nevertheless, the fact that the „cycling-like“ changes of prevalences of pathogens measured by SI_0 follows the (weak) clinical cycling is, in our opinion, quite striking. In this respect, what we presented is within-unit statistical significance. As mentioned, between-unit variability should be the next step. Our result is, as we mentioned in our manuscript, provisional. Our intention is to supply researchers with our method and see it applied to other records of antibiotic consumptions in order to see how other hospitals perform. 

- I think there is an important difference between the question presented in the introduction and in the literature (do mixing startegies reduce antibiotic resistance?) and the question mainly adressed in the results (do heterogeneity of treatment correlate with heterogeneity of resistance?). It is only in the last paragraph of the results that the authors attempt to adress the former.

Answer: We agree, the notations for different strategies have been used in a somewhat confusing way. Our aim is to provide a method that is capable to deal with diversity and with administrative strategies including cycling and mixing that may leave diversity invariant but are nevertheless relevant. Although mixing refers to a spatial permutation, this permutation takes place at a certain instance of time. This is why we carelessly generalised to refer to temporal changes. In our revised version we addressed the different strategies in a much more careful way. 

So in the current state, I do not see how the methods and the data presented in this article permit to adress what the authors present as the main question, and can not recommend the publication of this article as long as this is not clarified. But I am sure that there is potential interest in the data presented by the authors (time series of antibiotic treatments and pathogen resistance in different units of a hospital).

Below are more detailed comments about the manuscript, in three parts (A: abstract, B: introduction and methods, C: results).

A) I did not find the abstract understandable before reading the rest of the article:

"Temporal changes of the proportional abundances of different antibiotics (e.g. mixing or cycling)": unlike cycling, mixing does not implies temporal changes but spatial heterogeneity

"Although such a mixing strategy appears to be plausible": at this point mixing is not defined, and the previous sentence seems to rather relate to cycling than mixing ("temporal changes").

"We adopt diverse measures of heterogeneity and diversity": the authors should at least precise the variable whose they are trying to quantify heterogeneity of diversity

"We show that a reduction of prevalence of antibiotic-resistant germs correlates with a change of heterogeneity of antibiotics consumption" -> I would suggest to precise the direction of the change in the abstract

"we introduce a scheme based on linear regression for the assessment of associations between changes of heterogeneities on the antibiotics and the pathogen side" and "we show that a reduction of prevalence of antibiotic-resistant germs correlates with a change of heterogeneity of antibiotics consumption" -> I do not see what is new in the method (what does it mean to "introduce a linear regression scheme"? Linear regression is not new, and here is not really adapted for time-series data as explained above). All the authors do is plotting two curves next to each other and say they look similar (fig 13), without proper statistical analysis.

Answer: We addressed all points and substantially revised the abstract. With respect to objection „All the authors do is plotting two curves next to each other and say they look similar (fig 13), without proper statistical analysis.“ , our answer is that it depends a bit to which school of statisticians you belong. We did not just plot the curves next to each other, but we did a legitimate proper statistical inference based on confidence (overlapping CIs of slopes) rather than trusting too much in p-values. Anyhow, we now added a correlation analysis, but as expected, the inference does not change because the correlation is essentially the same as properly comparing the slopes in this case. 

B) The introduction and methods should be clarified, and need proper references to the literature:

L16 "whereas other strategies refer to a scheduled change of the dominantly used class of antibiotics": in the literature (including the one cited by the authors), antibiotic mixing does not involve scheduled change of antibiotic over time, but treatment of different subgroups of patients with different antibiotics.

Answer: We agree, as mentioned above, that the notations have been used in a confusing way. We revised accordingly. See also the above answers.

L17 "a fraction of patients": I think the authors should introduce the context better, to explain that these strategies are defined at the scale of a group of patients (for example in a hospital). This is important because with the definition provided by the authors, cycling could be wrongly understood as applied to individual patients.

Answer: We had „groups of patients“ in mind and revised the text accordingly.

L19 "Although there is some evidence that": citations needed. Overall there is a huge lack of references to the literature in the article.

Answer: We added references. 

L23-25 I find the sentence very unclear. What does it mean to "quantify the heterogeneity of [...] time courses of prevalence of antibiotic-resistant pathogens" ?

Answer: We changed to

„… quantify heterogeneity of both, antibiotic consumption as well as prevalence of antibiotic-resistant pathogens as a function of observation time.“

L57 is "DDDi" referring to the DDD or the "consumption density DDD"? If the former, why mentioning "consumption density DDD"?

Answer: We used DDD and added a corresponding clarification.

L56-57: At this point, it is not clear what is "the antibiotic group i". The previous paragraph explains that the hospital departments are clustered into 3 administrative units, and that the antibiotics are pooled in 12 antibiotic classes. To which kind of group this sentence refers to (are the authors computing antibiotic consumption for a class of antibiotic or for a unit of the hospital)? It only becomes clear much later in the article.

Answer: Antibiotic class was meant and we revised accordingly.

L60: Is the coefficient of variation computed for the vector of DDDi(t) for all i? Maybe clarify the formula?

Answer: We clarified that point: „… the mean is taken over the antibiotic classes and SD(DDD(t)) denotes the corresponding standard deviation.“

The paragraph "Coefficient of variation" must include references. It raises an interesting but rather simple point, and could be shortened.

Answer: We considerably reduced redundancy and added references to that paragraph.

The paragraph "Heterogeneity and entropy" is mostly paraphrasing the literature without proper citations. The authors do not explain how these theoretical ecology measures will be applied to quantify antibiotic resistance and antibiotic uses. I understand that they want to stay general because they will apply the same measures to different data, but everything would be more clear if the "antibiotic groups" had been properly defined (see my previous comment)

Answer: Intention was to address antibiotic classes and we revised accordingly. Please also see the related previous answer. 

L90, The definition of the "proportions" are not clear at this point. Do a1+a2+a3+...+an=1, or do ai+bi=1 for all i? Said otherwise, is "ai" the proportion of individuals of population a that belong to species i, or is it the proportion of individuals of species i that belong to environment a ? It only becomes clear much later in the text.

Answer: a_i is the proportion of individuals of population a that belong to species i (b analogues). We revised accordingly.

C) The data presented in the results (antibiotic consumption and resistance in a hospital) have some potential interest, but at this stage the authors do not show anything convincing.

Figure 1 : If I correctly understand, the fact that fig 1a and fig 1b look very similar suggest that the sum of DDD for all the antibiotic groups is almost constant over time. This should be made visible on Fig 1c by making the y-axis start at 0, otherwise a quick look at this panel conveys an inaccurate message.

Answer: We changed the scale of the y-axis accordingly.

L167-174 I do not see how what it presented in this paragraph is a result. The authors are mostly saying that the coefficient of variation is a good measure of variation.

Answer: With respect to the coefficient of variation (CV), the application belongs to a descriptive part. We discussed the application of CV mainly for the sake of completeness, for comparisons with other measures, and because CV is found in the literature as a measure of homogeneity. We occasionally refer to CV in the text and we think that it might be beneficial for some readers to see how the application of CV in such a case behaves. 

L 192-293 "Please note, the different scales of the y-axes reflect the different total amounts of consumption within each unit due to their different sizes": if this really comes from difference in size of the units (number of patients), Isn't it a strong argument for normalising DDD per patient days? I do not understand why the authors describe this normalisation in the methods but do not use it later in the results while it seems needed and appropriate.

Answer: All essential measures we discuss use proportions (relative abundances) as entry argument with the exception of CV. However, we find it illuminating to see the absolute DDDs. With respect to the results based on the diversity measures, it does not matter whether absolute DDDs or normalised DDDs are used. Relative abundances turn out to be almost identical for the two cases. We mentioned this but we did not show data due to redundancy. Moreover, the recorded patient days in the dataset may not be as reliable as the reports on antibiotic consumption. In addition, the surgical unit plays a uniqe role of treating mostly prophylactically. Residency in this unit per patient is short and patient days are not really meaningful. An absolute consumption of AB is arguably more informative.

L305 "The most important and at the same time most challenging question, in the given context, is whether the mixing behaviour of antibiotic administrations correlates or even causally relates to the prevalence of antibiotic resistances.": the authors do not show that their methods are useful for such a test. Do they really bring something compared to the coefficient of variation?

Answer: I do not see how the coefficient of variation can be modified such that a differential measure turns out comparable to SI_0. CV may reflect to some extent heterogeneity/diversity but it does – very much like HI - neither capture cycling nor mixing in an adequate way. However, HI (entropies) CAN be modified accordingly to yield SI_0. 

Figure 12 mentions "intensive care" and "normal care" unit, while everywhere else in the article the units are labeled "unit 1", "unit 2" and "unit 3"

Answer: Within the text we usually used the clear text in combination with the code to get an intuition of which unit we talk of. We introduced the code for labeling figs and the clear text for the fig legends, but unfortunately not in a consistent way. We revised the figures and text accordingly.

I do not understand the sentence "We conclude that mixing of antibiotic consumption correlates with the prevalence of antibiotic-resistant bacteria by means of parallel slopes of similarity indexes SI0"

Answer: This objection is perhaps due to our confusing usage of „mixing“ and „cycling“ (see previous answers). What the correlation of SI_0 for AB and SI_0 for pathogens show is, that the magnitude of effective (rather weak) long-term clinical cycling given by the slope of SI_0 is the same as the magnitude of effective cycling-like changes in pathogen prevalences. This might be trival, because reducing a certain AB may reduce the prevalence of pathogens resistant to that AB and vice versa. But the almost perfect correlation is nevertheless quite striking, in our opinion. Moreover, we have shown that the amount of „cycling“ expressed by SI_0 correlates strongly with a reduction of the prevalence of resistant bacteria. Please see previous answers as well as the substantially improved explanations throughout the revised text. 

Dataset: the authors made the effort of making the full dataset and analysis programs available. Just a minor suggestion: the variable names in R data sheets and the text in the R markdown file are currently in german, I think they would be more useful if translated to english.

Answer: We plan to upload the revised R-script to github upon acceptance of the manuscript.

---

## [Decision Letter · Decision Letter 1]

11 Aug 2020

PONE-D-19-25350R1

Mathematical basis for the assessment of antibiotic resistance and administrative counter-strategies

PLOS ONE

Dear Dr. Diebner,

Thank you for submitting your manuscript to PLOS ONE. After careful consideration, we feel that it has merit but does not fully meet PLOS ONE’s publication criteria as it currently stands. Therefore, we invite you to submit a revised version of the manuscript that addresses the points raised during the review process.

I appreciate your efforts in carefully addressing most of the points raised by both Reviewers. However, as you may see in their detailed comments below, although most of the points raised by Reviewer 2 have now been clarified, some concerns still remain about your time series statistical analysis. As a potential reader of your study, I also miss some more details on the specific statistical analysis that you conducted. This point is especially important for PLOS One publication criteria, available in the journal webpage.

We look forward to receiving your revised manuscript.

Kind regards,

Ricardo Martinez-Garcia

Academic Editor

PLOS ONE

Reviewers' comments:

Reviewer's Responses to Questions

**Comments to the Author**

1. If the authors have adequately addressed your comments raised in a previous round of review and you feel that this manuscript is now acceptable for publication, you may indicate that here to bypass the “Comments to the Author” section, enter your conflict of interest statement in the “Confidential to Editor” section, and submit your "Accept" recommendation.

Reviewer #2: (No Response)

2. Is the manuscript technically sound, and do the data support the conclusions?

Reviewer #2: Partly

3. Has the statistical analysis been performed appropriately and rigorously? 

Reviewer #2: I Don't Know

4. Have the authors made all data underlying the findings in their manuscript fully available?

Reviewer #2: Yes

5. Is the manuscript presented in an intelligible fashion and written in standard English?

Reviewer #2: Yes

6. Review Comments to the Author

Reviewer #2: Both in their answer and in the updated manuscript, Diebner and collaborators clarified that they see the mathematical framework as the main contribution of their article, and the application to the hospital dataset as a mere example of what can be achieved using this framework.

While most of my objections regarding the analysis of this hospital dataset and the interpretation of the findings remain, I guess they should thus not be seen as a blocking point. Besides, I think the authors clarified many points regarding their methodology and the context of their work. Many of my methodological questions are now addressed, in the manuscript or in the answer. As a biologist, it is still hard for me to judge the interest of the measures proposed by Diebner and colleagues without seeing a convincing biological application. So all I can say is that the methods and the context are clearly described, especially with the clarifications made in this revision, and that the figures are easily understandable.

Still, I am surprised that the authors do not see the problem with correlating two time-series which are both highly autocorrelated (L400-408). The statistical analysis is not described in details so I may be wrong, but I looked at the code and it seems that the authors directly compute a Pearson correlation between the two vectors (each vector being a time series). If so, I think this is quite inappropriate (because the data points of each vector are not independently sampled from a given distribution). This is what I was trying to point in my initial review by stating that ``only time series are analysed''. I have nothing per se against the analysis of time series, but it should be done correctly with the appropriate statistical tools (for example analysing the residuals).

7. PLOS authors have the option to publish the peer review history of their article (what does this mean?). If published, this will include your full peer review and any attached files.

Reviewer #2: No

---

## [Author Response · Author response to Decision Letter 1]

14 Aug 2020

Response to Reviewers

Manuscript PONE-D-19-25350R1

Reviewer #2:

Both in their answer and in the updated manuscript, Diebner and collaborators clarified that they see the mathematical framework as the main contribution of their article, and the application to the hospital dataset as a mere example of what can be achieved using this framework.

While most of my objections regarding the analysis of this hospital dataset and the interpretation of the findings remain, I guess they should thus not be seen as a blocking point. Besides, I think the authors clarified many points regarding their methodology and the context of their work. Many of my methodological questions are now addressed, in the manuscript or in the answer. As a biologist, it is still hard for me to judge the interest of the measures proposed by Diebner and colleagues without seeing a convincing biological application. So all I can say is that the methods and the context are clearly described, especially with the clarifications made in this revision, and that the figures are easily understandable.

Still, I am surprised that the authors do not see the problem with correlating two time-series which are both highly autocorrelated (L400-408). The statistical analysis is not described in details so I may be wrong, but I looked at the code and it seems that the authors directly compute a Pearson correlation between the two vectors (each vector being a time series). If so, I think this is quite inappropriate (because the data points of each vector are not independently sampled from a given distribution). This is what I was trying to point in my initial review by stating that ``only time series are analysed''. I have nothing per se against the analysis of time series, but it should be done correctly with the appropriate statistical tools (for example analysing the residuals).

Answer:

The reviewer raised an important point regarding the correlation of time series. Yes, strictly speaking, Pearson’s correlation requires independent sampling. Usually, the correlation of two time series is analysed by applying a cross-correlation-function which uses a time-lagged version of one of the time series. Unfortunately, this is not applicable here due to the low sampling frequency and very short time series. Thus, our argument here is that Pearson’s correlation is sufficient to at least raise hypotheses. Arguably, this supports the idea of a proof-of-principle. Based on the proposed analysis using diversity measures it is planned to design a proper study and we think that also other researchers dealing with antibiotic stewardship will benefit from our mathematical framework. We added the subsection “Statistical analysis” at the end of the “Materials and methods” section to elaborate on these points.

---

## [Editor Report · Decision Letter 2]

24 Aug 2020

Mathematical basis for the assessment of antibiotic resistance and administrative counter-strategies

PONE-D-19-25350R2

Dear Dr. Diebner,

We’re pleased to inform you that your manuscript has been judged scientifically suitable for publication and will be formally accepted for publication once it meets all outstanding technical requirements.

Kind regards,

Ricardo Martinez-Garcia

Academic Editor

PLOS ONE
---

## [Editor Report · Acceptance letter]

25 Aug 2020

PONE-D-19-25350R2 

Mathematical basis for the assessment of antibiotic resistance and administrative counter-strategies 

Dear Dr. Diebner:

I'm pleased to inform you that your manuscript has been deemed suitable for publication in PLOS ONE. Congratulations! Your manuscript is now with our production department. 

Kind regards, 

on behalf of

Dr. Ricardo Martinez-Garcia 

Academic Editor

PLOS ONE